# AUTOREGRESSIVE MOVING-AVERAGE ATTENTION MECHANISM FOR TIME SERIES FORECASTING

## ABSTRACT

We propose an Autoregressive (AR) Moving-average (MA) attention structure that can adapt to various linear attention mechanisms, enhancing their ability to capture long-range and local temporal patterns in time series. In this paper, we first demonstrate that, for the time series forecasting (TSF) task, the previously overlooked decoder-only autoregressive Transformer model can achieve results comparable to the best baselines when appropriate tokenization and training methods are applied. Moreover, inspired by the ARMA model from statistics and recent advances in linear attention, we introduce the full ARMA structure into existing autoregressive attention mechanisms. By using an indirect MA weight generation method, we incorporate the MA term while maintaining the time complexity and parameter size of the underlying efficient attention models. We further explore how indirect parameter generation can produce implicit MA weights that align with the modeling requirements for local temporal impacts. Experimental results show that incorporating the ARMA structure consistently improves the performance of various AR attentions on TSF tasks, achieving state-of-the-art results. The code implementation is available at the following underline{link}.

## 1 INTRODUCTION

In recent years, autoregressive (AR) decoder-only Transformer-based models (Vaswani, 2017; Radford, 2018) have been widely used in sequence modeling tasks across fields such as NLP (Brown et al., 2020; Touvron et al., 2023), CV (Chen et al., 2020; Esser et al., 2021; Chang et al., 2022), and audio (Borsos et al., 2023). This structure is well-suited for various sequential generation and prediction tasks. However, in typical sequence modeling tasks like time series forecasting (TSF), there has been less exploration of this architecture compared to other structures. Most of the best-performing recent TSF models are encoder-only Transformers (Liu et al., 2024a; Nie et al., 2022), MLPs (Das et al., 2023; Lu et al., 2024), or even linear models (Zeng et al., 2023; Xu et al., 2024). The few relevant discussions mainly focus on using pretrained autoregressive LLMs or similar structures for few-shot and zero-shot prediction (Gruver et al., 2023; Jin et al., 2024; Das et al., 2024; Liu et al., 2024b), with little research directly evaluating their TSF performance in end-to-end training. Therefore, this paper will first briefly demonstrate that with appropriate tokenization and training methods, a basic AR Transformer is enough to achieve results comparable to the state-of-the-art (SOTA) baselines, as shown in Fig. 1 and Fig. 2.

Recently, efficient linear AR attention variants have been explored and developed (Katharopoulos et al., 2020; Hua et al., 2022), reducing the time complexity of standard softmax attention from $O(N^2)$ to $O(N)$. Researchers have found that adding a gating decay factor or a similar exponential moving average (EMA) structure to AR structure, as in gated linear attention (Ma et al., 2022; Yang et al., 2024), enhances linear attention's ability to model local patterns and improves performance. The success of these approaches inspired us to introduce a more comprehensive full autoregressive moving-average (ARMA) structure into existing AR attention mechanisms and explore the performance of the ARMA Transformers in TSF.

In TSF models, EMA, connecting back to the historic work of Holt-Winters (Winters, 1960; Holt, 2004), focuses on smoothed local data, which improves the modeling of short-term fluctuations but reduces the ability to capture long-term information. In contrast, ARMA, connecting back to the historic work of Box-Jenkins (Box et al., 1974), a classic structure in TSF, considers both historical

data and the cumulative impact of prediction errors. This allows it to handle and decouple long-term and short-term effects, significantly improving forecasting performance on data with complicated temporal patterns.

We propose the ARMA attention mechanism, which integrates a moving-average (MA) term into various existing AR attention mechanisms. Our method improves the TSF performance of AR Transformers without significantly increasing computational costs, maintaining $O(N)$ time complexity and original parameter size. We design an indirect MA weight generation to obtain the MA output without explicitly computing the MA attention matrix, preserving efficiency of linear attentions. We explore specific techniques for generating implicit MA weights to ensure proper decoupling and handling of short-term effects. Extensive experiments and visualization analyses demonstrate that ARMA balances long- and short-term dependencies, significantly improving AR Transformers and achieving state-of-the-art TSF results.

The main contributions of this paper can be summarized as follows:

a) We demonstrate that, with appropriate tokenization and preprocessing methods, an AR Transformer is enough to achieve the level of existing SOTA baselines. Furthermore, the introduction of ARMA attention enables the decoder-only Transformer to outperform SOTA baselines.

b) We propose the ARMA attention mechanism, which introduces an MA term into existing AR attention without increasing time complexity or parameter size. By adding the MA term to various AR attention mechanisms, the resulting ARMA Transformers significantly improve forecasting performance compared to their AR counterparts.

c) We design an indirect MA weight generation method that is computationally efficient while ensuring that the implicit MA weights effectively capture the important short-term effects in TSF, allowing the AR term to focus more on long-term and cyclic patterns.

## 2 METHOD

### 2.1 TIME SERIES FORECASTING

In Time Series Forecasting (TSF), the goal is to predict the future part in a multivariate time series $\mathbf{S} \in \mathbb{R}^{L \times C}$, where $L$ is the length of the series, and $C$ is the number of channels or input series. The time series is divided into historical input $\mathbf{S}_I \in \mathbb{R}^{L_I \times C}$, and future data $\mathbf{S}_P \in \mathbb{R}^{L_P \times C}$, where $L = L_I + L_P$, and $L_I$ and $L_P$ represent the lengths of the input and forecasting periods, respectively. The objective is to learn a mapping function $f : \mathbb{R}^{L_I \times C} \to \mathbb{R}^{L_P \times C}$ that predicts the future values $\widehat{\mathbf{S}}_P = f(\mathbf{S}_I)$, given the historical input $\mathbf{S}_I$.

### 2.2 APPROPRIATE TOKENIZATION FOR AUTOREGRESSIVE FORECASTING

Recently, most time series forecasting research utilizes encoder-decoder or encoder-only Transformers for TSF (Li et al., 2019b; Zhou et al., 2021; Wu et al., 2021; Nie et al., 2022; Liu et al., 2024a), with limited focus on end-to-end decoder-only autoregressive Transformer because of error accumulation issue. For long-term forecasts, The autoregressive Transformers requires iteratively doing one-step prediction, leading to error accumulation and higher MSE compared to non-autoregressive models that generate the entire forecast at once.

To prevent error accumulation, we use an autoregressive Transformer (Fig. 1) that treats one-step prediction as the complete forecast. Inspired by PatchTST (Nie et al., 2022), we adopt a channel-independent approach, predicting each series separately and applying RevIN (Kim et al., 2022) to each. For an input series of length $L_I$ in $\mathbf{S}_I$, we apply non-overlapping patches with a patch size $L_P$, dividing the input into $N = \frac{L_I + P}{L_P}$ patches, where $P$ is zero-padding for divisibility. This ensures that each out-of-sample prediction token covers the entire forecasting length $L_P$, thereby avoiding error accumulation.

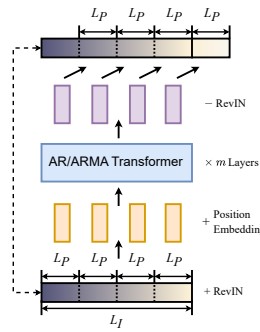

Figure 1: Overall architecture of our decoder Transformer for TSF.

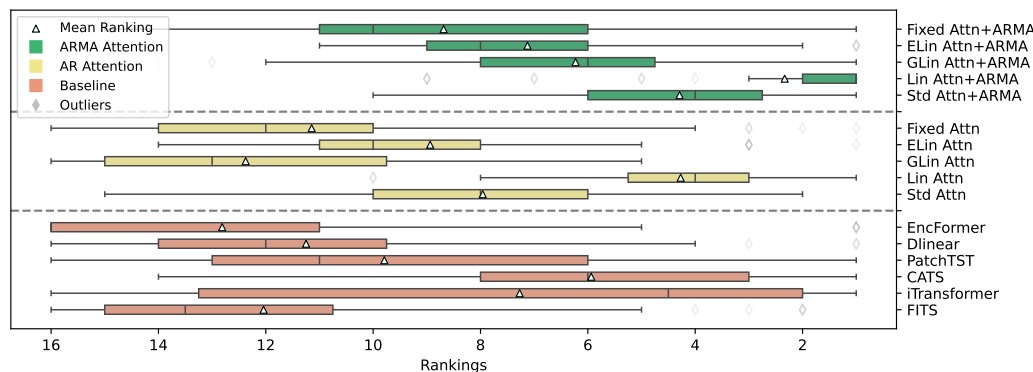

Figure 2: Box plots of performance rankings from 48 sub-experiments across 12 datasets. Green represents ARMA Transformers, yellow AR Transformers, and red the baselines, with triangles indicating mean rankings. AR Transformers perform comparably to baselines, while ARMA Transformers significantly outperform their AR counterparts. See Table 2 and 8 for more details.

Fig. 2 shows that autoregressive Transformers using this method can achieve performance comparable to existing SOTA models. Additionally, decoder-based architectures may have significant advantages in extended lookback length and varying output horizon, highlighting their potential.

## 2.3 PRELIMINARIES: DECODER-ONLY TRANSFORMER

We use a GPT-2–style decoder-only Transformer (Radford et al., 2019) for autoregressive TSF. Token patches of length $L_P$ are linearly projected to $d$-dimensional vectors and combined with learnable positional embeddings to form the input sequence $\mathbf{X} \in \mathbb{R}^{N \times d}$, where each token is $\boldsymbol{x}_t \in \mathbb{R}^{1 \times d}$. Each of the $m$ Transformer layers applies layer normalization $\text{LN}(\cdot)$, attention $\text{Attn}(\cdot)$, and a channel-wise MLP $\text{MLP}(\cdot)$. With single-head softmax attention, a Transformer layer is defined as:

$$\text{Attn}(\mathbf{X}) = \text{softmax}\left(\mathbf{M} \odot \left(\mathbf{Q}\mathbf{K}^\top\right)\right)\mathbf{V}\mathbf{W}_o, \text{ with } \mathbf{Q}, \mathbf{K}, \mathbf{V} = \mathbf{X}\mathbf{W}_q, \mathbf{X}\mathbf{W}_k, \mathbf{X}\mathbf{W}_v$$
$$\mathbf{X} := \mathbf{X} + \text{Attn}(\text{LN}(\mathbf{X})), \text{ then } \mathbf{X} := \mathbf{X} + \text{MLP}(\text{LN}(\mathbf{X})) \tag{1}$$

where $\mathbf{W}_q, \mathbf{W}_k, \mathbf{W}_v, \mathbf{W}_o \in \mathbb{R}^{d \times d}$ are the projection matrices for the query, key, value, and output, respectively, and $\mathbf{M} \in \mathbb{R}^{N \times N}$ is the causal mask, defined as $\mathbf{M}_{ij} = 1\{i \geq j\} - \infty \cdot 1\{i < j\}$.

## 2.4 PRELIMINARIES: EFFICIENT LINEAR ATTENTION MECHANISMS

Recent autoregressive efficient attention mechanisms reduce computational complexity from $O(N^2)$ to $O(N)$ by avoiding the explicit calculation of the $N \times N$ attention matrix (Katharopoulos et al., 2020; Choromanski et al., 2021; Hua et al., 2022; Sun et al., 2023). Most of them can be reformulated as parallel linear RNNs with identity or diagonal state updates. Although these efficient attentions do not outperform standard softmax attention for large models, they achieve comparable results on smaller tasks (Katharopoulos et al., 2020; Choromanski et al., 2021). This paper investigates integrating these mechanisms into TSF and shows that adding a moving-average term significantly improves their performance. We begin by expressing the recurrent form of standard softmax attention. For a single head without output projection, let $\boldsymbol{q}_t, \boldsymbol{k}_t, \boldsymbol{v}_t$ be the vectors at step $t$ from $\mathbf{Q}, \mathbf{K}, \mathbf{V}$. The output $\boldsymbol{o}_t$ is given by: $\boldsymbol{o}_t = \frac{\sum_{i=1}^{t} \exp(\boldsymbol{q}_t \boldsymbol{k}_i^\top) \boldsymbol{v}_i}{\sum_{i=1}^{t} \exp(\boldsymbol{q}_t \boldsymbol{k}_i^\top)}$.

**Linear attention** Linear Attention replaces the $\exp(\boldsymbol{q}_t \boldsymbol{k}_i^\top)$ term in standard attention with a kernel function $k(\boldsymbol{x}, \boldsymbol{y}) = \langle \phi(\boldsymbol{x}), \phi(\boldsymbol{y}) \rangle$, resulting in $\phi(\boldsymbol{q}_t)\phi(\boldsymbol{k}_i)$ (Katharopoulos et al., 2020). This change reduces the time complexity from $O(N^2)$ to $O(N)$ by eliminating the need to compute the full $N \times N$ attention matrix. Instead, it computes $\phi(\boldsymbol{k}_i)^\top \boldsymbol{v}_i$ for each $i$ and aggregates over $N$. Various kernel functions have been explored, with identity kernels without denominators performing well enough (Mao, 2022; Qin et al., 2022; Sun et al., 2023; Yang et al., 2024). In this setup, Linear attention can be viewed as an RNN with a hidden state matrix $\boldsymbol{k}_i^\top \boldsymbol{v}_i \in \mathbb{R}^{d \times d}$ that updates using the identity function. The output at each step is: $\boldsymbol{o}_t = \boldsymbol{q}_t \sum_{i=1}^{t} \boldsymbol{k}_i^\top \boldsymbol{v}_i$.

**Element-wise linear attention** In multi-head linear attention with $h$ heads, we handle $h$ hidden state matrices of size $\frac{d}{h} \times \frac{d}{h}$. When $h = d$, this simplifies to $h$ scalar hidden states, effectively transforming linear attention into a linear RNN with a $d$-dimensional hidden state vector $\phi(\boldsymbol{k}_i) \odot \boldsymbol{v}_i$ and enabling element-wise computations of $\boldsymbol{q}, \boldsymbol{k}, \boldsymbol{v}$. This approach, also known as the Attention Free Transformer (AFT) (Zhai et al., 2021), is favored for its simplicity and efficiency in recent works (Peng et al., 2023). We adopt the structure in AFT, where $\sigma(\cdot)$ is the sigmoid function, and the output at each step is: $\boldsymbol{o}_t = \sigma(\boldsymbol{q}_t) \odot \frac{\sum_{i=1}^{t} \exp(\boldsymbol{k}_i) \odot \boldsymbol{v}_i}{\sum_{i=1}^{t} \exp(\boldsymbol{k}_i)}$.

**Gated linear attention** Recent studies have explored adding a forget gate, commonly used in traditional RNNs, to linear attention, allowing autoregressive models to forget past information and focus on local patterns (Mao, 2022; Sun et al., 2023; Qin et al., 2024; Yang et al., 2024). We implement a simple gating mechanism where each input $\boldsymbol{x}_t$ is converted into a scalar between [0, 1] and expanded into a forget matrix $\mathbf{G}_i$ matching the shape of $\boldsymbol{k}_i \boldsymbol{v}_i$. With gating parameters $\mathbf{W}_g \in \mathbb{R}^{d \times 1}$, the output at each step is: $\boldsymbol{o}_t = \boldsymbol{q}_t \sum_{i=1}^{t} \mathbf{G}_i \odot \boldsymbol{k}_i^{\top} \boldsymbol{v}_i$, $\mathbf{G}_i = \prod_{k=1}^{i} \sigma(\boldsymbol{x}_k \mathbf{W}_g) \mathbf{1}^{\top} \mathbf{1}$.

**Fixed Attention** We additionally explore an autoregressive structure with fixed, data-independent weights $w_{t,i}$, replacing the dynamically generated attention weights $\phi(\boldsymbol{q}_t) \phi(\boldsymbol{k}_i)$. Without dynamic parameter generation, this becomes a linear layer with a causal mask $\mathbf{M}$ rather than a true attention mechanism. We use this structure to examine the effect of adding a moving-average term. This autoregressive causal linear layer is expressed as: $\boldsymbol{o}_t = \sum_{i=1}^{t} w_{t,i} \boldsymbol{v}_i$.

## 2.5 ARMA ATTENTION MECHANISM

In these attention mechanisms, the next-step prediction at time $t$ is a weighted sum of all previous values $\boldsymbol{v}_i \in \mathbb{R}^{1 \times d}$, with weights $\mathbf{w}_{t,i} \in \mathbb{R}^{1 \times d}$ derived from interactions between $\boldsymbol{q}_t$ and $\boldsymbol{k}_i$. Naturally, we can write these attention mechanisms in an AR model structure:

$$\boldsymbol{v}_{t+1} = \boldsymbol{o}_t^{\mathrm{AR}} + \boldsymbol{r}_t = \sum_{i=1}^{t} \mathbf{w}_{t,i} \odot \boldsymbol{v}_i + \boldsymbol{r}_t,$$

where $\boldsymbol{r}_t$ is the AR error. In an ARMA model, the MA term captures short-term fluctuations, allowing the AR component to focus on long-term dependencies. Let $\boldsymbol{\epsilon}_t$ be the error after introducing the MA term and $\boldsymbol{\theta}_{t-1,j}$ the MA weights generated by

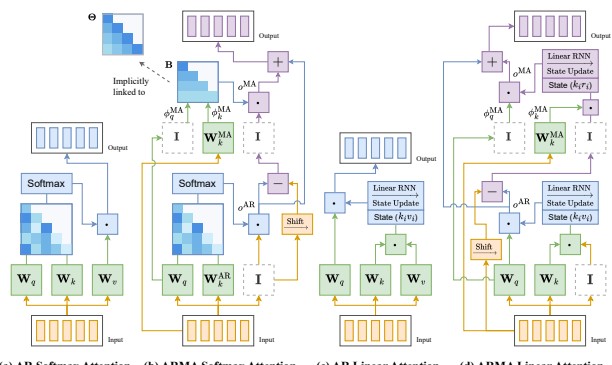

(a) AR Softmax Attention  (b) ARMA Softmax Attention  (c) AR Linear Attention  (d) ARMA Linear Attention

Figure 3: ARMA attention structure with the indirect MA weight generation method applied to softmax and linear attention. See Table 1 for more calculation details.

some attention mechanism. We expand the AR error $\boldsymbol{r}_t$ into an MA form and extend the model to an ARMA structure as:

$$\boldsymbol{v}_{t+1} = \boldsymbol{o}_t^{\mathrm{AR}} + \boldsymbol{o}_t^{\mathrm{MA}} + \boldsymbol{\epsilon}_t = \sum_{i=1}^{t} \mathbf{w}_{t,i} \odot \boldsymbol{v}_i + \sum_{j=1}^{t-1} \boldsymbol{\theta}_{t-1,j} \odot \boldsymbol{\epsilon}_j + \boldsymbol{\epsilon}_t, \quad \boldsymbol{r}_t = \sum_{j=1}^{t-1} \boldsymbol{\theta}_{t-1,j} \odot \boldsymbol{\epsilon}_j + \boldsymbol{\epsilon}_t \quad (2)$$

The structure of the MA output $\boldsymbol{o}_t^{\mathrm{MA}} = \sum_{j=1}^{t-1} \boldsymbol{\theta}_{t-1,j} \odot \boldsymbol{\epsilon}_j$ resembles the AR term and could potentially be computed using an attention mechanism. For simplicity, we consider a single channel of the $d$-dimensional space, with other channels an be handled in parallel. We express the matrix form of the $\boldsymbol{r}_t$ in Eq. 2 for one channel as:

$$\begin{pmatrix} r_1 \\ r_2 \\ \vdots \\ r_t \end{pmatrix} = \begin{pmatrix} 0 & 0 & \cdots & 0 & 0 \\ \theta_{1,1} & 0 & \cdots & 0 & 0 \\ \theta_{2,1} & \theta_{2,2} & \cdots & 0 & 0 \\ \vdots & \vdots & \ddots & \vdots & \vdots \\ \theta_{t-1,1} & \theta_{t-1,2} & \cdots & \theta_{t-1,t-1} & 0 \end{pmatrix} \begin{pmatrix} \epsilon_1 \\ \epsilon_2 \\ \vdots \\ \epsilon_t \end{pmatrix} + \begin{pmatrix} \epsilon_1 \\ \epsilon_2 \\ \vdots \\ \epsilon_t \end{pmatrix} \quad (3)$$

$$\mathbf{r} = (\mathbf{I} + \boldsymbol{\Theta})\boldsymbol{\epsilon}, \quad \boldsymbol{\epsilon} = (\mathbf{I} + \boldsymbol{\Theta})^{-1}\mathbf{r},$$

where $\boldsymbol{\Theta}$ is a strictly lower triangular matrix of MA weights for this channel. Once the attention mechanism determines $\boldsymbol{o}_t^{\text{AR}}$ and $\boldsymbol{\theta}_{t-1,j}$, we can calculate $\boldsymbol{r}_j = \boldsymbol{v}_{j+1} - \boldsymbol{o}_j^{\text{AR}}$ (token shifting) for all $j \leq N-1$ and determine $\boldsymbol{\epsilon}_j$ via matrix inversion. Substituting these back into Eq. 2 yields the final ARMA attention output $\boldsymbol{o}_t^{\text{AR}} + \boldsymbol{o}_t^{\text{MA}}$ for step $t$.

However, computing $\boldsymbol{o}_t^{\text{MA}}$ requires inverting $\mathbf{I} + \boldsymbol{\Theta}$, which involves calculating all $\boldsymbol{\theta}_{t-1,j}$ in the $N \times N$ matrix. This increases the complexity of linear attentions back to $O(N^2)$ and may also cause training instability. To maintain linear time complexity, we need a method to compute $\boldsymbol{o}_t^{\text{MA}}$ without explicitly calculating all $\boldsymbol{\theta}_{t-1,j}$ values.

## 2.6 INDIRECT MA WEIGHT GENERATION

We need an approach that can leverage linear attention's efficiency to compute $\boldsymbol{o}_t^{\text{MA}}$ without the costly $\boldsymbol{\Theta}^{N \times N}$ matrix operations. Instead of separately calculating attention weights to determine $\boldsymbol{\epsilon_j}$ as value input and recomputing the whole MA output, we aim to use a linear RNN to collect all keys and values at once. We observe from Eq. 3 that there is already a sequential relationship between $\boldsymbol{r}_j$ and $\boldsymbol{\epsilon}_j$, and $\boldsymbol{r}_j$ can be computed directly once $\boldsymbol{o}_t^{\text{AR}}$ is determined. Therefore, we implicitly compute the MA weights of $\boldsymbol{\epsilon_j}$ by using $\boldsymbol{r}_j$ as value input for the MA component instead of $\boldsymbol{\epsilon}_j$. Let $\boldsymbol{\beta}_{t-1,j}$ denote the generated attention weights corresponding to $\boldsymbol{r}_j$ at step $t$, and let $\boldsymbol{\theta}_{t-1,j}$ here be the implicit MA weights hiddenly linked to the generated $\boldsymbol{\beta}_{t-1,j}$. Based on Eq. 3, we establish:

$$\sum_{j=1}^{t-1} \boldsymbol{\beta}_{t-1,j} \odot \boldsymbol{r}_j = \sum_{j=1}^{t-1} \boldsymbol{\theta}_{t-1,j} \odot \boldsymbol{\epsilon}_j \Leftrightarrow \mathbf{B}\boldsymbol{r} = \boldsymbol{\Theta}\boldsymbol{\epsilon} \text{ (for one channel)}$$

$$\mathbf{B} = \boldsymbol{\Theta} \cdot (\mathbf{I} + \boldsymbol{\Theta})^{-1}, \quad \boldsymbol{\Theta} = \mathbf{B} \cdot (\mathbf{I} - \mathbf{B})^{-1}$$

(4)

With $\boldsymbol{\Theta} = \mathbf{B} \cdot (\mathbf{I} - \mathbf{B})^{-1}$, as long as the indirectly generated $\boldsymbol{\Theta}$ accurately reflects the characteristics of the MA weights we want, we can use $\sum_{j=1}^{t-1} \boldsymbol{\beta}_{t-1,j} \odot \boldsymbol{r}_j$ as $\boldsymbol{o}_t^{\text{MA}}$. Since $\boldsymbol{r}_j$ is known after computing $\boldsymbol{o}_t^{\text{AR}}$, linear attention can be used to compute $\boldsymbol{o}_t^{\text{MA}}$ without increasing the time complexity. To ensure the implicitly generated $\boldsymbol{\Theta}$ from $\mathbf{B}$ captures the desired MA properties, we must carefully design how $\mathbf{B}$ is generated. The invertibility of $(\mathbf{I} - \mathbf{B})^{-1}$ is guaranteed since $\mathbf{B}$ is strictly lower triangular. To efficiently compute the generated weights, we use the $\boldsymbol{\beta}_{t-1,j} = \phi_q^{\text{MA}}(\boldsymbol{q}_{t-1}^{\text{MA}})\phi_k^{\text{MA}}(\boldsymbol{k}_j^{\text{MA}})$ to generate $\mathbf{B}$, similar to linear attention. Previous dynamic ARMA models in statistics often update MA weights based on observations (Grenier, 1983; Azrak & Mélard, 2006), so we derive $\boldsymbol{q}_{t-1}^{\text{MA}}$ and $\boldsymbol{k}_j^{\text{MA}}$ by multiplying the attention input $\boldsymbol{x}$. with $\mathbf{W}_q^{\text{MA}}$ and $\mathbf{W}_k^{\text{MA}}$. Now, the effectiveness of MA weights lies in selecting the most suitable functions $\phi_q^{\text{MA}}(\cdot)$ and $\phi_k^{\text{MA}}(\cdot)$.

## 2.7 SELECTION OF $\phi(\cdot)$ AND CHARACTERISTICS OF IMPLICIT MA WEIGHTS

The MA term models short-term effects and local temporal relationships, so we want the implicit $\boldsymbol{\Theta}$ to follow a pattern where elements near the diagonal have larger absolute values, and those farther away gradually decrease. The expanded form of $\boldsymbol{\Theta}$ is given by $\boldsymbol{\Theta} = \mathbf{B} \cdot (\mathbf{I} - \mathbf{B})^{-1} = \mathbf{B} + \mathbf{B}^2 + \mathbf{B}^3 + \cdots$. The elements along the diagonal direction in $\mathbf{B}$ continually accumulate as products into the elements below them in $\boldsymbol{\Theta}$. Since $\mathbf{B}$ is strictly lower triangular, the elements of the subdiagonal in $\boldsymbol{\Theta}$ remain constant, while the elements further down progressively accumulate additional terms formed by the product of different $\beta$. elements above. Assuming $\beta$. follows a distribution and simplifying by setting each $\beta$. to the distribution mean $b$, the elements of $\boldsymbol{\Theta}$ can be expressed as:

$$\theta_{ij} = b(1+b)^{i-j-1}, \quad \text{where } i > j$$

(5)

This simplification offers valuable insights. To prevent longer-term errors from having a larger impact, we aim to avoid large absolute values accumulating in $\boldsymbol{\Theta}$ far from the diagonal. We also want $\theta$. to decay steadily as it moves away from the diagonal. Therefore, constraining $\beta$. between -1 and 0, with a preference of smaller absolute values, is a practical approach.

We tested various activation function combinations for $\phi_q^{\text{MA}}(\cdot)$ and $\phi_k^{\text{MA}}(\cdot)$ to generate $\boldsymbol{\beta}_{t-1,j} = \phi_q^{\text{MA}}(\boldsymbol{q}_{t-1}^{\text{MA}})\phi_k^{\text{MA}}(\boldsymbol{k}_j^{\text{MA}})$ values, as shown in Fig. 4, 7, and 8. We used the sigmoid function $\phi_k^{\text{MA}}(\boldsymbol{k}_j^{\text{MA}}) = \sigma(\alpha \boldsymbol{k}_j^{\text{MA}}/\sqrt{d})$ to obtain values between 0 and 1, where $\alpha = 0.05$ [1] and $\sqrt{d}$ are scaling

---

[1] In the key activation, $\alpha$ controls the variance of each row in the $\mathbf{B}$ matrix, indirectly influencing the amount of long-term information (lower left) in the MA weights $\boldsymbol{\Theta}$. Increasing $\alpha$ would make the MA weights focus

factors to maintain small absolute values. Then, we selected a function $\phi_q^{\text{MA}}(\cdot)$ to make the product negative. We ultimately chose $\phi_q^{\text{MA}}(\boldsymbol{q}_t^{\text{MA}}) = -\text{LeakyReLU}(-\boldsymbol{q}_t^{\text{MA}}/\sqrt{d})$ with a negative slope of 0.02. The inner negative sign maintains directional consistency (for later parameter sharing), and the outer negative sign encourages a negative output.

Fig. 4 shows that LeakyReLU provides a balanced lag weight pattern. Unlike ReLU and Sigmoid, which only output values of the same sign, LeakyReLU offers some relaxation while keeping most values negative. This adds flexibility by enabling the desired negative smoothing effect of the MA term, with occasional positive values to enhance modeling flexibility.

To summarize the ARMA attention process with indirect MA weight generation: First, we compute all $\boldsymbol{o}_t^{\text{AR}}$ using the selected attention mechanism. Then, we apply token shifting and compute all $\boldsymbol{r}_j$ for $j \leq N -$

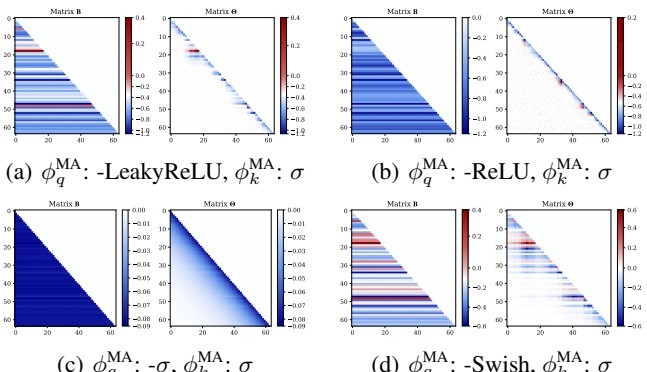

(a) $\phi_q^{\text{MA}}$: -LeakyReLU, $\phi_k^{\text{MA}}$: $\sigma$     (b) $\phi_q^{\text{MA}}$: -ReLU, $\phi_k^{\text{MA}}$: $\sigma$

(c) $\phi_q^{\text{MA}}$: -$\sigma$, $\phi_k^{\text{MA}}$: $\sigma$     (d) $\phi_q^{\text{MA}}$: -Swish, $\phi_k^{\text{MA}}$: $\sigma$

Figure 4: Visualization of the $\mathbf{B}$(left) $- \boldsymbol{\Theta}$(right) relationship with different $\phi(\cdot)$. We construct the simulated $\mathbf{B}$ matrices using randomly sampled $\boldsymbol{q}$ and $\boldsymbol{k}$ ($N = 64$, $d = 32$) from the normal distribution, and display the corresponding implicit $\boldsymbol{\Theta}$ matrices.

1. Next, using $\phi_q^{\text{MA}}(\boldsymbol{q}_t^{\text{MA}})$, $\phi_k^{\text{MA}}(\boldsymbol{k}_j^{\text{MA}})$, and $\boldsymbol{r}_j$, we calculate $\boldsymbol{o}_t^{\text{MA}}$ with the efficient method matching AR attention, as illustrated in Fig. 3. Finally, the ARMA output is $\boldsymbol{o}_t = (\boldsymbol{o}_t^{\text{AR}} + \boldsymbol{o}_t^{\text{MA}})\mathbf{W}_o$. A summary of MA computation methods for each attention mechanism is in Table 1.

**Computational cost and model performance** The introduction of MA term adds three weight matrices $\mathbf{W}_{\{q,k,v\}}^{\text{MA}}$, increasing parameter size. To ensure fair comparison, we use weight-sharing to match the parameter sizes of ARMA and AR models. Specifically, we share $\mathbf{W}_q$ between the AR and MA terms and set $\mathbf{W}_v$ to an identity matrix, with minimal impact due to the existance of $\mathbf{W}_o$ and the MLP layer (see Eq. 1). This reduces ARMA's trainable weights to $\mathbf{W}_q, \mathbf{W}_k^{\text{AR}}, \mathbf{W}_k^{\text{MA}}, \mathbf{W}_o$, as shown in Fig. 3. While ARMA attention has the same time complexity in order of magnitude as efficient AR attention, its two-stage structure may increase computational costs on constant level. We compare models with different number of layer in the experiments section to show that ARMA's improved performance is due to structural enhancements, but not increased complexity.

Table 1: Summary of ARMA attention for various attention mechanisms, detailing the calculation methods for AR output and MA output, where $\boldsymbol{r}_j = \boldsymbol{v}_{j+1} - \boldsymbol{o}_j^{\text{AR}}$.

| Model | AR term output $\boldsymbol{o}_t^{\text{AR}}$ | Indirect MA term output $\boldsymbol{o}_t^{\text{MA}}$ |
|---|---|---|
| Standard Softmax Attention (Std Attn) | $\frac{\sum_{i=1}^t \exp(\boldsymbol{q}_t(\boldsymbol{k}_i^{\text{AR}})^\top)\boldsymbol{v}_i}{\sum_{i=1}^t \exp(\boldsymbol{q}_t(\boldsymbol{k}_i^{\text{AR}})^\top)}$ | $\sum_{j=1}^{t-1} \phi_q^{\text{MA}}(\boldsymbol{q}_{t-1})\phi_k^{\text{MA}}(\boldsymbol{k}_j^{\text{MA}})^\top \boldsymbol{r}_j$ |
| Linear Attention (Lin Attn) | $\boldsymbol{q}_t \sum_{i=1}^t (\boldsymbol{k}_i^{\text{AR}})^\top \boldsymbol{v}_i$ | $\phi_q^{\text{MA}}(\boldsymbol{q}_{t-1}) \sum_{j=1}^{t-1} \phi_k^{\text{MA}}(\boldsymbol{k}_j^{\text{MA}})^\top \boldsymbol{r}_j$ |
| Element-wise Linear Attention (ELin Attn) | $\sigma(\boldsymbol{q}_t) \odot \frac{\sum_{i=1}^t \exp(\boldsymbol{k}_i^{\text{AR}}) \odot \boldsymbol{v}_i}{\sum_{i=1}^t \exp(\boldsymbol{k}_i^{\text{AR}})}$ | $\phi_q^{\text{MA}}(\boldsymbol{q}_{t-1}) \odot \sum_{j=1}^{t-1} \phi_k^{\text{MA}}(\boldsymbol{k}_j^{\text{MA}}) \odot \boldsymbol{r}_j$ |
| Gated Linear Attention (GLin Attn) | $\boldsymbol{q}_t \sum_{i=1}^t \mathbf{G}_i \odot (\boldsymbol{k}_i^{\text{AR}})^\top \boldsymbol{v}_i$ | $\phi_q^{\text{MA}}(\boldsymbol{q}_{t-1}) \sum_{j=1}^{t-1} \phi_k^{\text{MA}}(\boldsymbol{k}_j^{\text{MA}})^\top \boldsymbol{r}_j$ |
| Fixed Attention (Fixed Attn) | $\sum_{i=1}^t w_{t,i}^{\text{AR}} \boldsymbol{v}_i$ | $\phi_q^{\text{MA}}(\boldsymbol{w}_{t-1}^{\text{MA,q}}) \sum_{j=1}^{t-1} \phi_k^{\text{MA}}(\boldsymbol{w}_j^{\text{MA,k}})^\top \boldsymbol{r}_j$ |

# 3 EXPERIMENTS

We conducted comprehensive experiments on 12 widely-used TSF datasets, including Weather, Solar, Electricity (ECL), ETTs, Traffic, and PEMS. See §A.1 for detailed description of datasets.

---

more on modeling long-term information. However, since we want the AR weights to handle the long-term component, we set $\alpha$ to a relatively small value. This explains why the rows of the $\mathbf{B}$ matrix appear smooth in the visualization. Refer to Fig. 7 for more details on $\alpha$, and see Fig. 8 for the effects of reversed positive $\phi_q$.

Table 2: Summary of main TSF results with forecasting horizons $L_P \in \{12, 24, 48, 96\}$ and $L_I = 512$. See Table 8 for the original results. Averages of test set MSE for each model on each dataset are presented. Average rankings (AvgRank) of each model, along with the count of first-place rankings (#Top1), are also included. Green indicates better performance, while red indicates worse. For each comparison between AR and ARMA attention, the better model is underlined.

| Model | AR/ARMA Transformer | | | | | | | | | | Baseline | | | | | |
|---|---|---|---|---|---|---|---|---|---|---|---|---|---|---|---|---|
| | Std Attn | Std Attn +ARMA | Lin Attn | Lin Attn +ARMA | GLin Attn | GLin Attn +ARMA | ELin Attn | ELin Attn +ARMA | Fixed Attn | Fixed Attn +ARMA | FITS | iTransformer | CATS | PatchTST | DLinear | Enc-Former |
| Weather | 0.104 | 0.101 | 0.104 | 0.100 | 0.119 | 0.105 | 0.104 | 0.103 | 0.105 | 0.104 | 0.114 | 0.117 | 0.105 | 0.107 | 0.124 | 0.135 |
| Solar | 0.134 | 0.124 | 0.122 | 0.119 | 0.148 | 0.124 | 0.136 | 0.133 | 0.142 | 0.135 | 0.152 | 0.145 | 0.122 | 0.150 | 0.149 | 0.125 |
| ECL | 0.110 | 0.106 | 0.106 | 0.104 | 0.110 | 0.108 | 0.115 | 0.114 | 0.121 | 0.118 | 0.124 | 0.106 | 0.110 | 0.111 | 0.114 | 0.201 |
| ETTh1 | 0.323 | 0.318 | 0.318 | 0.316 | 0.408 | 0.321 | 0.323 | 0.321 | 0.330 | 0.328 | 0.333 | 0.351 | 0.327 | 0.335 | 0.329 | 0.817 |
| ETTh2 | 0.192 | 0.192 | 0.193 | 0.195 | 0.217 | 0.198 | 0.193 | 0.190 | 0.200 | 0.194 | 0.197 | 0.229 | 0.194 | 0.201 | 0.198 | 0.597 |
| ETTm1 | 0.264 | 0.239 | 0.238 | 0.222 | 0.407 | 0.260 | 0.246 | 0.244 | 0.267 | 0.251 | 0.237 | 0.259 | 0.222 | 0.244 | 0.235 | 0.429 |
| ETTm2 | 0.131 | 0.128 | 0.126 | 0.121 | 0.142 | 0.128 | 0.134 | 0.128 | 0.129 | 0.127 | 0.115 | 0.135 | 0.116 | 0.119 | 0.120 | 0.311 |
| Traffic | 0.341 | 0.333 | 0.337 | 0.330 | 0.429 | 0.350 | 0.352 | 0.348 | 0.373 | 0.365 | 0.385 | 0.330 | 0.372 | 0.358 | 0.375 | 0.847 |
| PEMS03 | 0.112 | 0.100 | 0.100 | 0.096 | 0.209 | 0.101 | 0.116 | 0.112 | 0.121 | 0.116 | 0.133 | 0.096 | 0.105 | 0.140 | 0.134 | 0.111 |
| PEMS04 | 0.118 | 0.106 | 0.103 | 0.098 | 0.167 | 0.105 | 0.122 | 0.119 | 0.128 | 0.124 | 0.151 | 0.098 | 0.108 | 0.164 | 0.148 | 0.099 |
| PEMS07 | 0.092 | 0.083 | 0.087 | 0.077 | 0.093 | 0.087 | 0.101 | 0.097 | 0.106 | 0.100 | 0.132 | 0.079 | 0.094 | 0.093 | 0.129 | 0.102 |
| PEMS08 | 0.148 | 0.132 | 0.119 | 0.116 | 0.159 | 0.125 | 0.150 | 0.144 | 0.161 | 0.152 | 0.201 | 0.117 | 0.135 | 0.121 | 0.193 | 0.183 |
| AvgRank | 7.958 | 4.292 | 4.271 | 2.333 | 12.375 | 6.229 | 8.938 | 7.125 | 11.146 | 8.688 | 12.042 | 7.271 | 5.938 | 9.792 | 11.250 | 12.813 |
| #Top1 | 0 | 4 | 4 | 25 | 0 | 1 | 1 | 3 | 1 | 3 | 0 | 5 | 4 | 1 | 2 | 4 |

**Baselines** We built AR Transformers using the five attention mechanisms from Table 1 and added MA terms to create ARMA attention for comparison in TSF tasks. Additionally, we included five recent SOTA baselines: FITS (Xu et al., 2024), iTransformer (Liu et al., 2024a), CATS (Lu et al., 2024), PatchTST (Nie et al., 2022), and DLinear (Zeng et al., 2023). We also used a simple channel-dependent encoder-only Transformer, modified by repeating the last input value (like NLinear) to address distribution shift. This model already surpasses older architectures like Autoformer (Wu et al., 2021) and Informer (Zhou et al., 2021), so we excluded these from our comparison.

In the main experiments, both AR and ARMA Transformers use a consistent setup: $m = 3$ Transformer layers, 8 heads, and model dimension determined by a empirical method $d = 16\sqrt{C}$, where $C$ is the number of series. We evaluate their performance using one-step prediction for each test datapoint, aligned with the baselines. Baseline hyperparameters are set to the reported values from their original papers. For more details on hyperparameters and implementation, see §A.2.

$L_I$, $L_P$, **and fair comparison** AR/ARMA Transformers maintain stable performance across different input lengths $L_I$ and forecasting horizons $L_P$. In the main experiments, we set $L_I = 512$ and test with $L_P \in \{12, 24, 48, 96\}$. We did not select longer $L_P$ values for the **main experiments** because the token length in our AR/ARMA Transformer is determined by $L_I/L_P$. For longer horizons like $L_P = 720$, AR/ARMA Transformers need a larger $L_I$ (e.g., 4096). If $L_I = 512$ is used, it results in only one token, reducing the model to a simple MLP, which is feasible but fails to demonstrate the strengths of the AR/ARMA structure. However, increasing $L_I$ for baselines would hurt their performance, as $L_I = 512$ is within the optimal range for them. To ensure fairness, we use these settings for the main experiments and later include additional experiments extending $L_I$ to 4096 and $L_P$ to 720.

We ran all models on all datasets for the four different $L_P$. In the main text, we report the average test set MSE for each model across different $L_P$ on each dataset and provide the full results in §A.3.

**Main TSF results** Table 2 highlights the significant performance gains from extending AR Transformers to the ARMA structure. All ARMA attention mechanisms outperform their AR counterparts in both average test MSE and ranking, with linear and standard attention showing the best results.

**Performance of linear attention** Linear attention outperforms softmax attention in TSF, suggesting that simpler attention patterns and non-normalized input shortcuts (without denominator) can improve generalization on time-varying distributions. This aligns with earlier findings where linear models can outperform more complex Transformers in TSF (Zeng et al., 2023; Xu et al., 2024).

**Performance of gated linear attention** ARMA brought the greatest improvement to gated linear attention. In gated AR models, the decay factor helps the AR term focus on important local patterns, but it weakens the ability to capture long-term or stable cyclic patterns. By introducing the MA term, local effects are absorbed, allowing the decay factor to function properly in the AR forgetting mechanism, leading to significant performance gains.

**Performance of fixed attention** Fixed attention, which lacks dynamic parameter generation, performs worse than other attention. However, its significant improvement with MA terms shows that ARMA enhances the model's ability structurally to capture comprehensive sequence patterns.

Table 3: Summary showing that ARMA Transformers with $m = 3$ layers consistently outperform their AR counterparts across a wide range of $m$. The same experimental settings and data presentation method as in Table 2 are used. See Table 9 for the original results.

| | Model | $m=3$ ARMA | $m=1$ AR | $m=2$ AR | $m=3$ AR | $m=4$ AR | $m=5$ AR | $m=6$ AR | $m=7$ AR | $m=8$ AR |
|---|---|---|---|---|---|---|---|---|---|---|
| Weather | Std Attn | 0.101 | 0.109 | 0.108 | 0.104 | 0.108 | 0.113 | 0.111 | 0.113 | 0.112 |
| | Lin Attn | 0.100 | 0.104 | 0.103 | 0.104 | 0.103 | 0.103 | 0.103 | 0.102 | 0.103 |
| | GLin Attn | 0.105 | 0.122 | 0.122 | 0.119 | 0.121 | 0.121 | 0.122 | 0.121 | 0.120 |
| | ELin Attn | 0.103 | 0.110 | 0.107 | 0.104 | 0.108 | 0.109 | 0.111 | 0.110 | 0.111 |
| | Fixed Attn | 0.104 | 0.113 | 0.109 | 0.105 | 0.110 | 0.112 | 0.110 | 0.110 | 0.110 |
| ETTm1 | Std Attn | 0.239 | 0.265 | 0.270 | 0.264 | 0.266 | 0.269 | 0.270 | 0.270 | 0.272 |
| | Lin Attn | 0.222 | 0.241 | 0.233 | 0.238 | 0.232 | 0.230 | 0.230 | 0.231 | 0.231 |
| | GLin Attn | 0.260 | 0.411 | 0.413 | 0.407 | 0.409 | 0.410 | 0.410 | 0.409 | 0.404 |
| | ELin Attn | 0.244 | 0.253 | 0.251 | 0.246 | 0.253 | 0.257 | 0.259 | 0.256 | 0.258 |
| | Fixed Attn | 0.251 | 0.269 | 0.264 | 0.267 | 0.260 | 0.258 | 0.259 | 0.258 | 0.257 |

**Performance and complexity** The improvement of adding the MA term comes from its ability to model short-term impacts, allowing the AR term to focus on long-term and cyclic effects, not from increased computational costs. Table 3 shows that, regardless of the number of layers $m$ (1 to 8), AR Transformers consistently underperform compared to ARMA Transformers with a fixed $m = 3$.

Table 4: Summary showing that AR/ARMA Transformers effectively utilize extended lookback $L_I$, while baselines experience performance degradation. $L_I \in \{512, 1024, 2048, 4096\}$ with $L_P \in \{12, 24, 48, 96\}$ are evaluated and averaged. Original results can be found in Table 10.

| | Model | AR/ARMA Transformer | | | | | | | | | | Baseline | | | | | |
|---|---|---|---|---|---|---|---|---|---|---|---|---|---|---|---|---|---|
| | | Std Attn | Std Attn +ARMA | Lin Attn | Lin Attn +ARMA | GLin Attn | GLin Attn +ARMA | ELin Attn | ELin Attn +ARMA | Fixed Attn | Fixed Attn +ARMA | FITS | iTransformer | CATS | PatchTST | DLinear | EncFormer |
| Weather | $L_I=512$ | 0.104 | 0.101 | 0.104 | 0.100 | 0.119 | 0.105 | 0.104 | 0.103 | 0.105 | 0.104 | 0.114 | 0.117 | 0.105 | 0.108 | 0.124 | 0.135 |
| | $L_I=1024$ | 0.107 | 0.102 | 0.102 | 0.101 | 0.116 | 0.104 | 0.106 | 0.106 | 0.108 | 0.105 | 0.120 | 0.117 | 0.108 | 0.120 | 0.118 | 0.124 |
| | $L_I=2048$ | 0.110 | 0.102 | 0.101 | 0.100 | 0.114 | 0.102 | 0.108 | 0.108 | 0.123 | 0.110 | 0.121 | 0.119 | 0.113 | 0.122 | 0.119 | 0.128 |
| | $L_I=4096$ | 0.108 | 0.102 | 0.100 | 0.100 | 0.115 | 0.105 | 0.109 | 0.107 | 0.110 | 0.108 | 0.124 | 0.132 | 0.123 | 0.125 | 0.121 | 0.136 |
| ETTm1 | $L_I=512$ | 0.264 | 0.239 | 0.238 | 0.222 | 0.407 | 0.260 | 0.246 | 0.244 | 0.267 | 0.251 | 0.237 | 0.259 | 0.222 | 0.244 | 0.235 | 0.429 |
| | $L_I=1024$ | 0.280 | 0.241 | 0.239 | 0.227 | 0.423 | 0.236 | 0.265 | 0.253 | 0.281 | 0.263 | 0.240 | 0.258 | 0.238 | 0.245 | 0.239 | 0.364 |
| | $L_I=2048$ | 0.278 | 0.239 | 0.233 | 0.223 | 0.327 | 0.232 | 0.281 | 0.252 | 0.288 | 0.268 | 0.246 | 0.248 | 0.261 | 0.250 | 0.239 | 0.415 |
| | $L_I=4096$ | 0.275 | 0.234 | 0.237 | 0.226 | 0.324 | 0.229 | 0.282 | 0.265 | 0.287 | 0.266 | 0.252 | 0.274 | 0.340 | 0.260 | 0.250 | 0.428 |

**Adaptability to Longer $L_I$** Previous baseline models typically use $L_I$ between 96 and 720, as longer $L_I$ often leads to overfitting to long-term patterns, ignoring more important local effects (Zeng et al., 2023; Nie et al., 2022; Liu et al., 2024a). However, the next-step prediction and varying lookback inputs in AR/ARMA Transformers help the model focus on tokens closer to the next step, improving generalization. As shown in Table 4, increasing $L_I$ from 512 to 4096 improves AR/ARMA performance, demonstrating scalability and the ability to properly leverage long-term effects. Also, the ARMA structure consistently boosts AR model performance across different $L_I$.

**Varying $L_P$** We evaluated AR/ARMA Transformers with $L_I = 4096$ on long-term output $L_P = 720$ while keeping the best-performing $L_I = 512$ for the baselines. We also compared short-term output with $L_P = 1$ using $L_I = 512$ for all models. As shown in Table 5, across different $L_P$, AR models performed comparably to baselines, and the ARMA structure consistently improved performance over AR models.

Table 5: Summary of model performance on varying horizons $L_P$. AR/ARMA Transformers uses $L_I = 512$ for $L_P = 1$ and $L_I = 4096$ for $L_P = 720$. Baselines are consistently set to their best-performing $L_I = 512$ configuration. Original results can be found in Table 11.

| | Model | AR/ARMA Transformer | | | | | | | | | | Baseline | | | | | |
|---|---|---|---|---|---|---|---|---|---|---|---|---|---|---|---|---|---|
| | | Std Attn | Std Attn +ARMA | Lin Attn | Lin Attn +ARMA | GLin Attn | GLin Attn +ARMA | ELin Attn | ELin Attn +ARMA | Fixed Attn | Fixed Attn +ARMA | FITS | iTransformer | CATS | PatchTST | DLinear | EncFormer |
| Weather | $L_P=1$ | 0.037 | 0.032 | 0.032 | 0.031 | 0.033 | 0.031 | 0.037 | 0.034 | 0.036 | 0.034 | 0.038 | 0.038 | 0.033 | 0.035 | 0.034 | 0.032 |
| | $L_P=720$ | 0.299 | 0.301 | 0.308 | 0.305 | 0.310 | 0.299 | 0.296 | 0.295 | 0.299 | 0.297 | 0.327 | 0.327 | 0.311 | 0.325 | 0.319 | 0.404 |
| ETTm1 | $L_P=1$ | 0.048 | 0.043 | 0.042 | 0.041 | 0.051 | 0.043 | 0.053 | 0.051 | 0.054 | 0.052 | 0.047 | 0.049 | 0.043 | 0.046 | 0.044 | 0.044 |
| | $L_P=720$ | 0.396 | 0.391 | 0.417 | 0.408 | 0.396 | 0.393 | 0.395 | 0.391 | 0.403 | 0.399 | 0.420 | 0.438 | 0.418 | 0.408 | 0.420 | 0.802 |
| ETTm2 | $L_P=1$ | 0.034 | 0.031 | 0.032 | 0.030 | 0.033 | 0.030 | 0.035 | 0.033 | 0.033 | 0.031 | 0.034 | 0.034 | 0.032 | 0.033 | 0.031 | 0.035 |
| | $L_P=720$ | 0.329 | 0.327 | 0.341 | 0.338 | 0.334 | 0.328 | 0.327 | 0.326 | 0.331 | 0.327 | 0.360 | 0.369 | 0.351 | 0.362 | 0.403 | 2.641 |

**Comparison to MEGA** The MEGA structure (Ma et al., 2022) uses an exponential moving average (EMA) in gated attention to model local patterns. However, applying EMA directly to AR weights weakens the model's ability to capture long-term and stable seasonal patterns, making it less effective than ARMA at decoupling long-term and short-term effects. Table 6 shows that the performance of using MEGA as the attention mechanism is similar to using gated linear attention without the MA term. It provides less improvement compared to gated linear attention with ARMA.

**Visualization analysis** Fig. 5 shows test loss curves for different AR/ARMA attention mechanisms on the Weather and ETTm1 datasets, with ARMA consistently outperforming AR in both

Table 6: Summary of the performance comparison with MEGA. See Table 12 for the original result.

| Model | Std Attn | Std Attn +ARMA | Lin Attn | Lin Attn +ARMA | GLin Attn | GLin Attn +ARMA | MEGA |
|---|---|---|---|---|---|---|---|
| Weather | 0.104 | 0.101 | 0.104 | 0.100 | 0.119 | 0.105 | 0.121 |
| Solar | 0.134 | 0.124 | 0.122 | 0.119 | 0.148 | 0.124 | 0.226 |
| ETTh1 | 0.323 | 0.318 | 0.318 | 0.316 | 0.408 | 0.321 | 0.404 |
| ETTh2 | 0.192 | 0.192 | 0.193 | 0.195 | 0.217 | 0.198 | 0.214 |
| ETTm1 | 0.264 | 0.239 | 0.238 | 0.222 | 0.407 | 0.260 | 0.412 |
| ETTm2 | 0.131 | 0.128 | 0.126 | 0.121 | 0.142 | 0.128 | 0.137 |
| PEMS03 | 0.112 | 0.100 | 0.100 | 0.096 | 0.209 | 0.101 | 0.161 |

convergence speed and final loss. Fig. 6 visualizes attention input sequence, AR weights, $\mathbf{B}$, and $\Theta$ matrices of a test datapoint on Weather, showing how MA weights decouple local patterns, allowing AR weights to focus on cyclic and long-term patterns. Additional visualizations in Figs. 9–12 reinforce that there are important long-term stable seasonal patterns for AR weights to capture that should not be disrupted by applying forget gates or EMA. This explains why gated linear attention underperforms linear attention in our experiments.

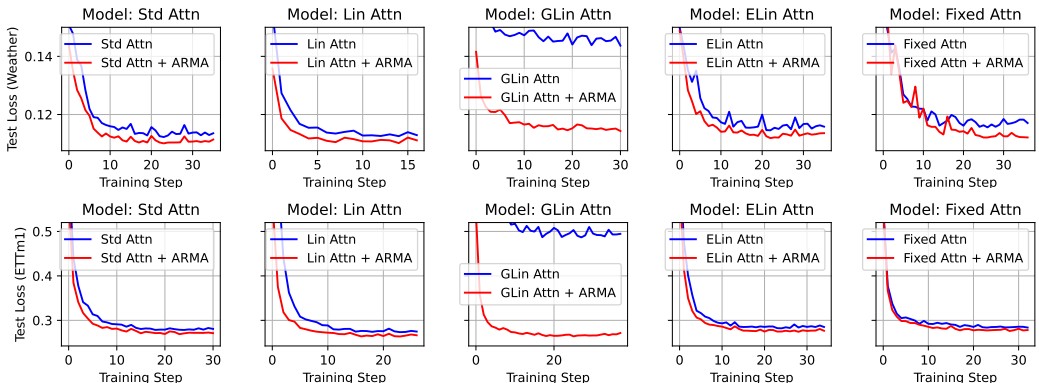

Figure 5: Visualization of test loss curves. We show the testing performance of five attention mechanisms using AR/ARMA structures on the Weather and ETTm1 datasets ($L_I = 512$, $L_P = 48$).

**Computational cost** Table 7 compares the computational cost of AR/ARMA Transformers with baselines on the ETTm1 dataset. Our tokenization method reduces the token size $N$, keeping AR/ARMA models' computational cost comparable to the baselines. Additionally, parameter sharing ensures the MA term doesn't increase the number of parameters, and the extra FLOPs from using ARMA are not significant.

Table 7: Comparison of computational costs utilizing the data format of ETTm1 to build model inputs ($L_I = 512$). The hyper-parameters for models are set according to their default configurations.

| Models | EncFormer | | CATS | | PatchTST | | iTransformer | | DLinear | | FITS | |
|---|---|---|---|---|---|---|---|---|---|---|---|---|
| Metric | FLOPs | Params | FLOPs | Params | FLOPs | Params | FLOPs | Params | FLOPs | Params | FLOPs | Params |
| $L_P = 96$ | 1.442G | 1.646M | 262.9M | 1.326M | 180.9M | 1.046M | 81.96M | 1.857M | 4.337M | 98.50K | 334.0K | 24.02K |
| $L_P = 48$ | 1.328G | 1.646M | 243.5M | 1.227M | 163.6M | 652.9K | 81.69M | 1.851M | 2.174M | 49.25K | 308.1K | 22.16K |
| $L_P = 24$ | 1.271G | 1.645M | 233.9M | 1.178M | 155.0M | 456.3K | 81.56M | 1.848M | 1.093M | 24.62K | 294.2K | 21.16K |
| $L_P = 12$ | 1.242G | 1.645M | 229.0M | 1.154M | 150.7M | 358.0K | 81.49M | 1.847M | 552.5K | 12.31K | 288.3K | 20.74K |

| Model | GLin Attn | | GLin Attn +ARMA | | Lin Attn | | Lin Attn +ARMA | | ELin Attn | | ELin Attn +ARMA | |
|---|---|---|---|---|---|---|---|---|---|---|---|---|
| Metric | FLOPs | Params | FLOPs | Params | FLOPs | Params | FLOPs | Params | FLOPs | Params | FLOPs | Params |
| $L_P = 96$ | 7.403M | 45.81K | 7.431M | 45.81K | 7.387M | 45.79K | 7.415M | 45.79K | 7.258M | 45.79K | 7.266M | 45.79K |
| $L_P = 48$ | 12.63M | 43.97K | 12.77M | 43.97K | 12.60M | 43.95K | 12.74M | 43.95K | 12.36M | 43.95K | 12.37M | 43.95K |
| $L_P = 24$ | 24.30M | 45.22K | 24.70M | 45.22K | 24.25M | 45.21K | 24.64M | 45.21K | 23.77M | 45.21K | 23.80M | 45.21K |
| $L_P = 12$ | 46.58M | 49.82K | 47.45M | 49.82K | 46.46M | 49.80K | 47.34M | 49.80K | 45.54M | 49.80K | 45.60M | 49.80K |

## 4 RELATED WORKS

**Linear attention with exponential moving-average** Beyond using a gating decay factor on the hidden state matrix of linear attention (Mao, 2022; Sun et al., 2023; Yang et al., 2024), recent studies have explored incorporating EMA mechanisms into gated linear attention by applying a smoothing factor (summing to 1) to the two terms in the state update (Ma et al., 2022). Similar EMA mechanisms have also been used in many modern RNN structures (Gu et al., 2022; Peng et al., 2023; Orvieto et al., 2023; Qin et al., 2024). Additionally, Schiele et al. (2022) attempted to introduce the ARMA structure into traditional RNNs, but their method could not ensure that the generated MA

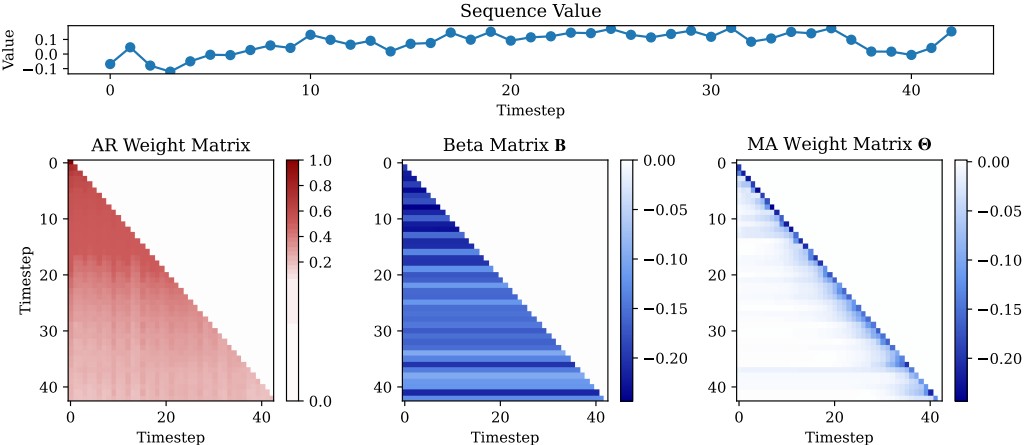

Figure 6: Visualization of the ARMA attention weights (first attention layer, averaged across the multiple heads or $d$-dimensional channels) for the first test set data point in the Weather dataset ($L_I = 4096, L_P = 96$). More ARMA weight visualization can be found in Fig. 9, 10, 11, and 12.

weights can properly model short-term patterns, and the final results did not significantly surpass traditional RNNs nor compare with recent attention models.

**TSF Structures** The use of neural network structures for TSF has been widely explored (Hochreiter & Schmidhuber, 1997; Rangapuram et al., 2018; Salinas et al., 2020). Recently, many Transformer-based TSF models with encoder-only and encoder-decoder structures have emerged (Li et al., 2019a; Zhou et al., 2021; Wu et al., 2021; Zhang & Yan, 2023; Nie et al., 2022; Liu et al., 2024a). However, these complex Transformer architectures have not significantly outperformed simpler MLP or linear models (Zeng et al., 2023; Das et al., 2023; Xu et al., 2024; Lu et al., 2024). Additionally, these models struggle to handle short-term effects properly with longer lookback windows, where, paradoxically, longer inputs often lead to worse performance.

## 5 CONCLUSION, LIMITATION, AND FUTURE WORKS

We propose the ARMA attention mechanism, which integrates an MA term into existing AR attention using a novel indirect MA weight generation method. This approach maintains the same time complexity and parameter size while ensuring the validity of the implicit MA weights. Experiments demonstrate that ARMA attention successfully decouples and handles long-term and short-term effects. The ARMA Transformer, enhanced with the MA term, outperforms their AR counterparts and achieves state-of-the-art results, offering consistent improvements in training with minimal added computational cost.

One limitation is that we have not explored combining the channel-independent ARMA Transformer with multivariate forecasting models to improve its handling of inter-series relationships. For future work, ARMA attention could be applied to general sequence modeling tasks beyond TSF. Testing on larger-scale datasets, such as using ARMA Transformers for large-scale NLP pretraining, is another promising direction.

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

# A APPENDIX

## A.1 DATASETS

Our main MTSF experiments are conducted on 12 widely-used real-world time series datasets. These datasets are summarized as follows:

**Weather Dataset**[1]**(Wu et al., 2021)** comprises 21 meteorological variables, including air temperature and humidity, recorded at 10-minute intervals throughout 2020 from the Weather Station of the Max Planck Biogeochemistry Institute in Germany.

**Solar Dataset**[2]**(Lai et al., 2018)** consists of high-frequency solar power production data from 137 photovoltaic plants recorded throughout 2006. Samples were collected at 10-minute intervals.

**Electricity Dataset**[3]**(Wu et al., 2021)** contains hourly electricity consumption records for 321 consumers over a three-year period from 2012 to 2014.

**ETT Dataset**[4]**(Zhou et al., 2021)** The ETT (Electricity Transformer Temperature) Dataset comprises load and oil temperature data from two electricity transformers, recorded at 15-minute and hourly intervals from July 2016 to July 2018. It is divided into four subsets (ETTm1, ETTm2, ETTh1, and ETTh2), each containing seven features related to oil and load characteristics.

**Traffic Dataset**[5]**(Wu et al., 2021)** Sourced from 862 freeway sensors in the San Francisco Bay area, the Traffic dataset provides hourly road occupancy rates from January 2015 to December 2016. This comprehensive dataset offers consistent measurements across a two-year period.

**PEMS Dataset**[6]**(Li et al., 2017)** The PEMS dataset consists of public traffic network data collected in California at 5-minute intervals. Our study utilizes four widely-adopted subsets (PEMS03, PEMS04, PEMS07, and PEMS08), which have been extensively studied in the field of spatial-temporal time series analysis for traffic prediction tasks.

---

[1] `https://www.bgc-jena.mpg.de/wetter/`
[2] `http://www.nrel.gov/grid/solar-power-data.html`
[3] `https://archive.ics.uci.edu/ml/datasets/ElectricityLoadDiagrams20112014`
[4] `https://github.com/zhouhaoyi/ETDataset`
[5] `http://pems.dot.ca.gov/`
[6] `http://pems.dot.ca.gov/`

## A.2 Hyper-parameter settings and implementation details

For the hyper-parameter settings of the AR/ARMA Transformer, we use $m = 3$ Transformer layers, 8 heads, and set the hidden dimension $d$ based on the number of series $C$, using the empirical formula $d = 16\sqrt{C}$. We use $4d$ as the hidden dimension for the feedforward MLP in the Transformer layer. A dropout rate of 0.1 is applied to both the AR term and MA term. We initialize the weights of all linear layers and embedding layers using the GPT-2 weight initialization method, with a normal distribution and a standard deviation of 0.02. For the output projection layers in the attention and MLP, we additionally scale the standard deviation by a factor of $1/\sqrt{m}$, aligned with the GPT-2 setting. Normalization layer is applied both before the input to the Transformer and after the Transformer output. We experimented with both standard LayerNorm and RMSNorm as the normalization layer, finding no significant performance differences, so we opted for RMSNorm for lower computational cost. For token input projection, we use a linear layer to project the $L_P$-dimensional token to a $d$-dimensional input vector. In the output projection, we do not tie the weights between the input and output linear layers. A learnable position embedding that maps the integer labels from 1 to $N$ (the input sequence length) to the corresponding $d$-dimensional position vectors is used. At the beginning of the model, we apply RevIN to input series $\mathbf{S}_I$, subtracting the mean and dividing by the standard deviation for each series. Before outputting the final result, we multiply by the standard deviation and add the mean back. All input series are processed independently and in parallel, merging different series dimensions into the batch size for parallel computation. The random seed used in all the experiments is 2024.

All training tasks in this paper can be conducted using a single Nvidia RTX 4090 GPU. The batch size is set to 32. For larger datasets, such as Traffic and PEMS07, we use a batch size of 16 or 8, with 2-step or 4-step gradient accumulation to ensure the effective batch size for parameter updates remains 32. During training, AR/ARMA Transformers are trained using the next-step prediction objective with MSE loss. We use the AdamW optimizer with betas=(0.9, 0.95) and weight decay=0.1, following the GPT-2 settings. For a fair comparison, the same optimizer is used for training baseline models. It is important to note that the baseline models trained with this AdamW setup show significantly better TSF performance compared to those trained with the default Adam optimizer settings. As a result, the baseline performance presented in this paper may exceed the results reported in their original papers. Since this study focuses on long-term last token prediction results, we apply an additional weight factor to the training loss for the last token, multiplying it by $N$. However, this weighting only slightly affects performance on smaller datasets with fewer data points, such as ETTs, and has little to no effect on larger datasets. Given the minimal impact of this method, the original next-token MSE loss is sufficient for most datasets, without requiring further modifications.

We use the same train-validation-test set splitting ratio as in previous studies by Zeng et al. (2023); Nie et al. (2022); Liu et al. (2024a). We also follow the same dataset standardization methods used in these studies. During training, we evaluate the validation and test losses at the end of each epoch, with an early-stopping patience set to 12 epochs. The maximum number of training epochs is 100. We apply a linear warm-up for the learning rate, increasing it from 0.00006 to 0.0006 over the first 5 epochs, and gradually decreasing it in the subsequent epochs.

## A.3 Supplementary experiment results

In the following section, we provide the complete experimental data corresponding to the tables in the main text. Additionally, we include extra visualizations to help illustrate the actual behavior of the MA weights.

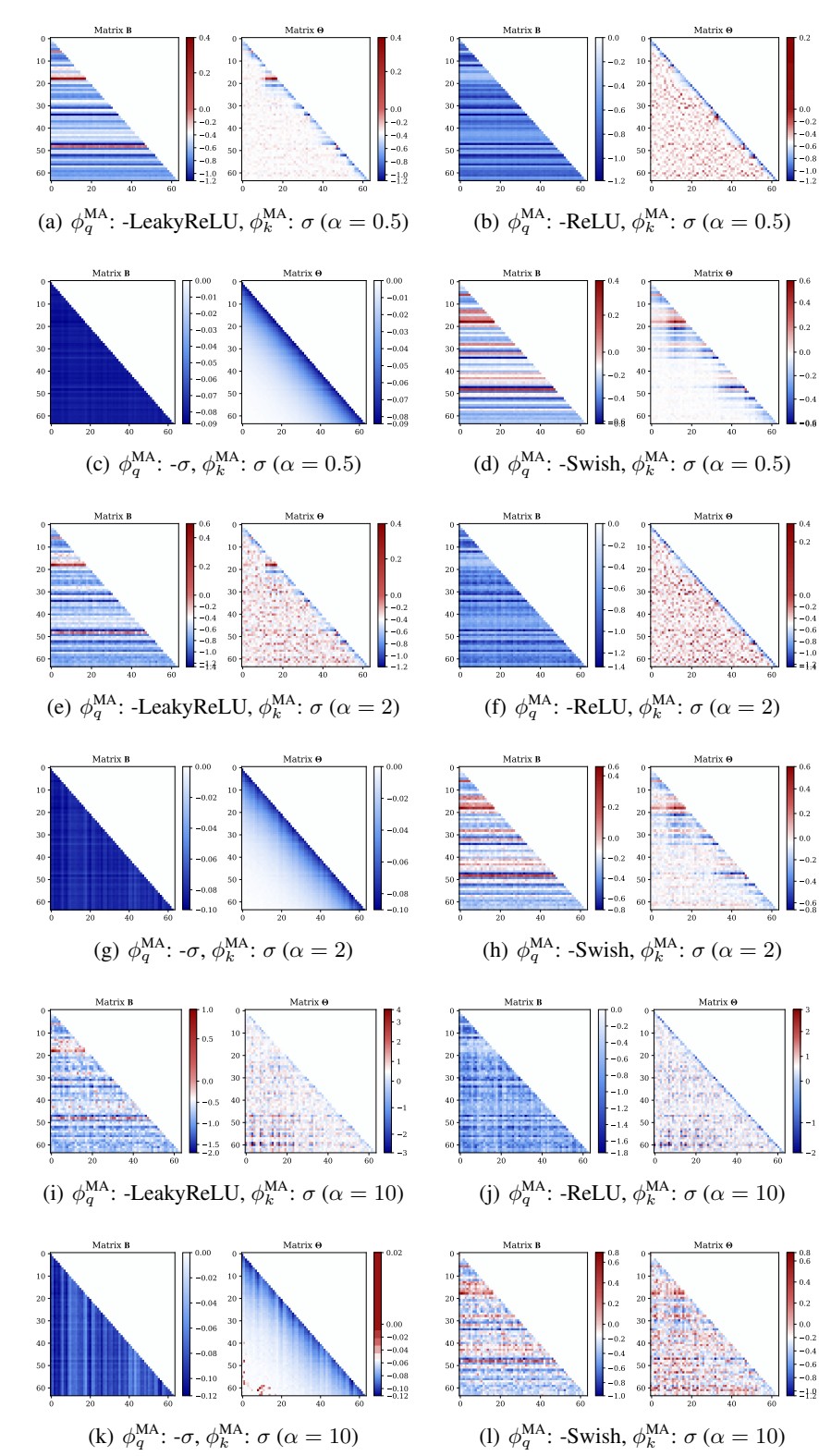

Figure 7: Additional visualization of $\mathbf{B} - \boldsymbol{\Theta}$ relationship with different $\phi(\cdot)$ and different $\alpha$. We construct the simulated $\mathbf{B}$ matrices using randomly sampled $\boldsymbol{q}$ and $\boldsymbol{k}$ ($N = 64$, $d = 32$) from the normal distribution, and display the corresponding $\boldsymbol{\Theta}$ matrices.

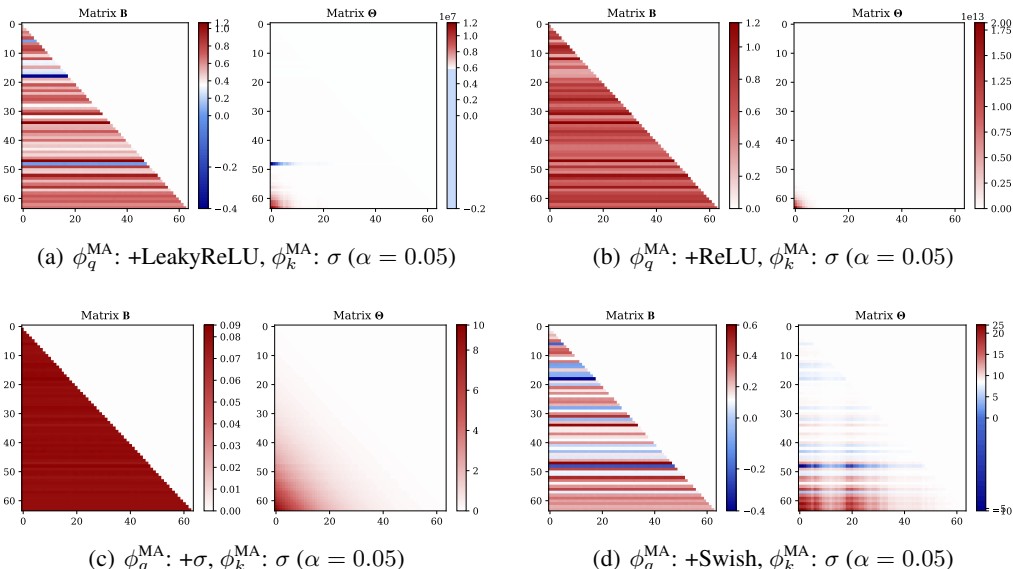

Figure 8: Visualization of $\mathbf{B} - \boldsymbol{\Theta}$ relationship with positive query activation functions $\phi_q^{\text{MA}}(\cdot)$.

Table 8: Detailed results of main TSF experiments with forecasting horizons $L_P \in \{12, 24, 48, 96\}$ and $L_I = 512$. Test set MSE and MAE for each model on each experiment setup are presented.

| Model | Metrics | Std Attn | | Std Attn +ARMA | | Lin Attn | | Lin Attn +ARMA | | GLin Attn | | GLin Attn +ARMA | | ELin Attn | | ELin Attn +ARMA | | Fixed Attn | | Fixed Attn +ARMA | | FITS | | iTransformer | | CATS | | PatchTST | | DLinear | | EncFormer | |
|---|---|---|---|---|---|---|---|---|---|---|---|---|---|---|---|---|---|---|---|---|---|---|---|---|---|---|---|---|---|---|---|---|---|---|
| | | MSE | MAE | MSE | MAE | MSE | MAE | MSE | MAE | MSE | MAE | MSE | MAE | MSE | MAE | MSE | MAE | MSE | MAE | MSE | MAE | MSE | MAE | MSE | MAE | MSE | MAE | MSE | MAE | MSE | MAE | MSE | MAE |
| Weather | 96 | 0.144 | 0.195 | 0.142 | 0.193 | 0.142 | 0.194 | 0.139 | 0.191 | 0.161 | 0.210 | 0.142 | 0.194 | 0.146 | 0.197 | 0.143 | 0.195 | 0.147 | 0.194 | 0.142 | 0.198 | 0.151 | 0.204 | 0.158 | 0.140 | 0.146 | 0.198 | 0.149 | 0.224 | 0.150 | 0.209 | 0.188 | 0.248 |
| | 48 | 0.113 | 0.157 | 0.109 | 0.151 | 0.115 | 0.158 | 0.110 | 0.153 | 0.144 | 0.191 | 0.116 | 0.159 | 0.114 | 0.158 | 0.112 | 0.156 | 0.112 | 0.155 | 0.115 | 0.159 | 0.125 | 0.177 | 0.132 | 0.176 | 0.116 | 0.160 | 0.118 | 0.161 | 0.122 | 0.177 | 0.143 | 0.199 |
| | 24 | 0.089 | 0.118 | 0.085 | 0.115 | 0.088 | 0.117 | 0.083 | 0.114 | 0.101 | 0.129 | 0.090 | 0.122 | 0.087 | 0.116 | 0.089 | 0.119 | 0.091 | 0.122 | 0.088 | 0.120 | 0.103 | 0.147 | 0.101 | 0.139 | 0.090 | 0.125 | 0.092 | 0.122 | 0.147 | 0.102 | 0.117 | 0.162 |
| | 12 | 0.071 | 0.091 | 0.067 | 0.086 | 0.069 | 0.088 | 0.067 | 0.086 | 0.070 | 0.090 | 0.070 | 0.091 | 0.069 | 0.087 | 0.069 | 0.087 | 0.071 | 0.089 | 0.069 | 0.090 | 0.078 | 0.112 | 0.078 | 0.105 | 0.069 | 0.092 | 0.070 | 0.097 | 0.078 | 0.115 | 0.093 | 0.120 |
| Solar | 96 | 0.196 | 0.263 | 0.192 | 0.257 | 0.183 | 0.247 | 0.180 | 0.244 | 0.209 | 0.283 | 0.182 | 0.243 | 0.194 | 0.258 | 0.191 | 0.257 | 0.195 | 0.261 | 0.187 | 0.256 | 0.210 | 0.254 | 0.230 | 0.257 | 0.182 | 0.239 | 0.209 | 0.251 | 0.208 | 0.274 | 0.201 | 0.227 |
| | 48 | 0.160 | 0.230 | 0.151 | 0.223 | 0.152 | 0.219 | 0.149 | 0.217 | 0.177 | 0.258 | 0.154 | 0.222 | 0.162 | 0.236 | 0.159 | 0.233 | 0.161 | 0.233 | 0.159 | 0.229 | 0.188 | 0.245 | 0.189 | 0.220 | 0.157 | 0.214 | 0.186 | 0.240 | 0.184 | 0.379 | 0.157 | 0.186 |
| | 24 | 0.112 | 0.180 | 0.098 | 0.168 | 0.098 | 0.166 | 0.095 | 0.162 | 0.143 | 0.232 | 0.099 | 0.167 | 0.115 | 0.192 | 0.113 | 0.188 | 0.131 | 0.221 | 0.122 | 0.203 | 0.132 | 0.289 | 0.108 | 0.163 | 0.097 | 0.161 | 0.129 | 0.203 | 0.128 | 0.208 | 0.092 | 0.290 |
| | 12 | 0.069 | 0.137 | 0.055 | 0.118 | 0.056 | 0.113 | 0.052 | 0.111 | 0.063 | 0.139 | 0.059 | 0.121 | 0.072 | 0.135 | 0.069 | 0.133 | 0.081 | 0.161 | 0.070 | 0.141 | 0.078 | 0.159 | 0.053 | 0.104 | 0.052 | 0.113 | 0.075 | 0.155 | 0.075 | 0.161 | 0.048 | 0.405 |
| ECL | 96 | 0.136 | 0.233 | 0.132 | 0.229 | 0.130 | 0.226 | 0.128 | 0.225 | 0.135 | 0.231 | 0.133 | 0.229 | 0.139 | 0.237 | 0.138 | 0.235 | 0.142 | 0.239 | 0.139 | 0.236 | 0.144 | 0.246 | 0.132 | 0.226 | 0.131 | 0.229 | 0.132 | 0.224 | 0.135 | 0.232 | 0.232 | 0.342 |
| | 48 | 0.117 | 0.214 | 0.113 | 0.211 | 0.110 | 0.207 | 0.110 | 0.206 | 0.113 | 0.210 | 0.113 | 0.210 | 0.124 | 0.223 | 0.121 | 0.219 | 0.125 | 0.223 | 0.121 | 0.220 | 0.129 | 0.233 | 0.111 | 0.207 | 0.115 | 0.214 | 0.112 | 0.206 | 0.120 | 0.219 | 0.198 | 0.317 |
| | 24 | 0.097 | 0.193 | 0.094 | 0.190 | 0.092 | 0.188 | 0.101 | 0.189 | 0.099 | 0.197 | 0.096 | 0.193 | 0.103 | 0.202 | 0.100 | 0.199 | 0.106 | 0.205 | 0.104 | 0.202 | 0.115 | 0.221 | 0.095 | 0.191 | 0.101 | 0.199 | 0.103 | 0.196 | 0.106 | 0.206 | 0.183 | 0.302 |
| | 12 | 0.089 | 0.188 | 0.085 | 0.184 | 0.091 | 0.191 | 0.088 | 0.189 | 0.091 | 0.190 | 0.088 | 0.188 | 0.093 | 0.197 | 0.096 | 0.201 | 0.110 | 0.218 | 0.107 | 0.215 | 0.106 | 0.214 | 0.084 | 0.186 | 0.091 | 0.192 | 0.098 | 0.199 | 0.096 | 0.196 | 0.194 | 0.313 |
| ETTh1 | 96 | 0.357 | 0.393 | 0.360 | 0.395 | 0.358 | 0.396 | 0.361 | 0.399 | 0.378 | 0.406 | 0.368 | 0.404 | 0.360 | 0.393 | 0.356 | 0.393 | 0.362 | 0.394 | 0.359 | 0.394 | 0.369 | 0.398 | 0.396 | 0.422 | 0.371 | 0.398 | 0.386 | 0.407 | 0.370 | 0.394 | 0.986 | 0.720 |
| | 48 | 0.334 | 0.375 | 0.331 | 0.374 | 0.331 | 0.375 | 0.331 | 0.376 | 0.349 | 0.386 | 0.337 | 0.381 | 0.330 | 0.372 | 0.331 | 0.375 | 0.331 | 0.373 | 0.334 | 0.378 | 0.343 | 0.379 | 0.362 | 0.396 | 0.341 | 0.379 | 0.352 | 0.389 | 0.342 | 0.379 | 0.884 | 0.670 |
| | 24 | 0.312 | 0.365 | 0.299 | 0.357 | 0.299 | 0.356 | 0.299 | 0.357 | 0.349 | 0.394 | 0.303 | 0.360 | 0.305 | 0.360 | 0.305 | 0.360 | 0.306 | 0.361 | 0.304 | 0.360 | 0.318 | 0.366 | 0.334 | 0.381 | 0.313 | 0.365 | 0.313 | 0.361 | 0.312 | 0.362 | 0.792 | 0.676 |
| | 12 | 0.290 | 0.345 | 0.280 | 0.340 | 0.285 | 0.342 | 0.272 | 0.337 | 0.554 | 0.503 | 0.277 | 0.340 | 0.296 | 0.351 | 0.348 | 0.348 | 0.320 | 0.368 | 0.316 | 0.368 | 0.301 | 0.357 | 0.310 | 0.363 | 0.281 | 0.341 | 0.290 | 0.345 | 0.291 | 0.348 | 0.607 | 0.534 |
| ETTh2 | 96 | 0.266 | 0.330 | 0.268 | 0.331 | 0.273 | 0.336 | 0.275 | 0.338 | 0.285 | 0.338 | 0.281 | 0.345 | 0.267 | 0.333 | 0.263 | 0.329 | 0.288 | 0.345 | 0.276 | 0.337 | 0.270 | 0.336 | 0.373 | 0.394 | 0.270 | 0.338 | 0.274 | 0.341 | 0.277 | 0.346 | 0.568 | 0.596 |
| | 48 | 0.213 | 0.290 | 0.216 | 0.294 | 0.215 | 0.290 | 0.217 | 0.293 | 0.233 | 0.386 | 0.220 | 0.295 | 0.212 | 0.290 | 0.210 | 0.288 | 0.215 | 0.250 | 0.208 | 0.288 | 0.217 | 0.298 | 0.226 | 0.310 | 0.212 | 0.299 | 0.222 | 0.304 | 0.217 | 0.301 | 0.475 | 0.434 |
| | 24 | 0.160 | 0.252 | 0.160 | 0.252 | 0.159 | 0.250 | 0.162 | 0.252 | 0.182 | 0.263 | 0.164 | 0.254 | 0.162 | 0.249 | 0.158 | 0.249 | 0.158 | 0.250 | 0.158 | 0.251 | 0.168 | 0.264 | 0.178 | 0.275 | 0.167 | 0.262 | 0.174 | 0.269 | 0.166 | 0.263 | 0.290 | 0.405 |
| | 12 | 0.129 | 0.229 | 0.125 | 0.224 | 0.124 | 0.224 | 0.125 | 0.224 | 0.168 | 0.263 | 0.127 | 0.224 | 0.129 | 0.228 | 0.128 | 0.230 | 0.139 | 0.240 | 0.133 | 0.235 | 0.133 | 0.239 | 0.139 | 0.248 | 0.128 | 0.235 | 0.135 | 0.242 | 0.131 | 0.237 | 0.225 | 0.354 |
| ETTm1 | 96 | 0.305 | 0.359 | 0.301 | 0.354 | 0.303 | 0.355 | 0.296 | 0.351 | 0.336 | 0.382 | 0.299 | 0.355 | 0.305 | 0.356 | 0.301 | 0.354 | 0.298 | 0.347 | 0.296 | 0.344 | 0.305 | 0.347 | 0.352 | 0.378 | 0.300 | 0.354 | 0.323 | 0.362 | 0.305 | 0.348 | 0.686 | 0.603 |
| | 48 | 0.287 | 0.344 | 0.276 | 0.333 | 0.278 | 0.336 | 0.266 | 0.328 | 0.500 | 0.472 | 0.282 | 0.334 | 0.284 | 0.346 | 0.282 | 0.342 | 0.284 | 0.338 | 0.280 | 0.340 | 0.280 | 0.331 | 0.304 | 0.346 | 0.266 | 0.328 | 0.289 | 0.341 | 0.278 | 0.329 | 0.475 | 0.474 |
| | 24 | 0.246 | 0.312 | 0.223 | 0.293 | 0.218 | 0.293 | 0.196 | 0.279 | 0.473 | 0.429 | 0.225 | 0.305 | 0.239 | 0.305 | 0.241 | 0.307 | 0.284 | 0.332 | 0.255 | 0.321 | 0.218 | 0.289 | 0.223 | 0.293 | 0.197 | 0.277 | 0.235 | 0.304 | 0.216 | 0.288 | 0.352 | 0.391 |
| | 12 | 0.218 | 0.287 | 0.156 | 0.240 | 0.151 | 0.240 | 0.128 | 0.222 | 0.320 | 0.335 | 0.144 | 0.237 | 0.157 | 0.247 | 0.153 | 0.246 | 0.203 | 0.282 | 0.261 | 0.261 | 0.144 | 0.235 | 0.155 | 0.246 | 0.125 | 0.222 | 0.128 | 0.224 | 0.140 | 0.230 | 0.204 | 0.294 |
| ETTm2 | 96 | 0.177 | 0.262 | 0.174 | 0.261 | 0.167 | 0.255 | 0.162 | 0.250 | 0.178 | 0.260 | 0.172 | 0.258 | 0.178 | 0.260 | 0.174 | 0.260 | 0.167 | 0.254 | 0.166 | 0.254 | 0.164 | 0.253 | 0.198 | 0.277 | 0.168 | 0.255 | 0.177 | 0.266 | 0.184 | 0.283 | 0.481 | 0.525 |
| | 48 | 0.139 | 0.238 | 0.137 | 0.236 | 0.145 | 0.248 | 0.143 | 0.250 | 0.167 | 0.264 | 0.148 | 0.249 | 0.153 | 0.253 | 0.139 | 0.238 | 0.146 | 0.245 | 0.138 | 0.238 | 0.126 | 0.225 | 0.147 | 0.242 | 0.127 | 0.227 | 0.131 | 0.231 | 0.125 | 0.226 | 0.483 | 0.516 |
| | 24 | 0.121 | 0.219 | 0.117 | 0.217 | 0.110 | 0.211 | 0.101 | 0.211 | 0.132 | 0.227 | 0.119 | 0.209 | 0.117 | 0.216 | 0.116 | 0.216 | 0.119 | 0.220 | 0.107 | 0.221 | 0.096 | 0.195 | 0.112 | 0.212 | 0.096 | 0.195 | 0.108 | 0.194 | 0.095 | 0.194 | 0.162 | 0.280 |
| | 12 | 0.086 | 0.175 | 0.083 | 0.174 | 0.083 | 0.174 | 0.078 | 0.169 | 0.089 | 0.178 | 0.082 | 0.172 | 0.086 | 0.174 | 0.083 | 0.174 | 0.085 | 0.177 | 0.083 | 0.174 | 0.073 | 0.168 | 0.082 | 0.181 | 0.072 | 0.165 | 0.072 | 0.165 | 0.077 | 0.195 | 0.116 | 0.231 |
| Traffic | 96 | 0.379 | 0.273 | 0.373 | 0.269 | 0.365 | 0.262 | 0.362 | 0.260 | 0.474 | 0.324 | 0.381 | 0.275 | 0.381 | 0.270 | 0.378 | 0.271 | 0.393 | 0.393 | 0.389 | 0.277 | 0.404 | 0.286 | 0.356 | 0.259 | 0.377 | 0.284 | 0.381 | 0.298 | 0.399 | 0.286 | 0.915 | 0.460 |
| | 48 | 0.352 | 0.256 | 0.342 | 0.253 | 0.349 | 0.251 | 0.339 | 0.254 | 0.569 | 0.371 | 0.330 | 0.263 | 0.363 | 0.263 | 0.330 | 0.261 | 0.374 | 0.275 | 0.365 | 0.265 | 0.393 | 0.286 | 0.341 | 0.264 | 0.369 | 0.271 | 0.373 | 0.275 | 0.385 | 0.284 | 0.857 | 0.434 |
| | 24 | 0.324 | 0.238 | 0.315 | 0.231 | 0.322 | 0.239 | 0.318 | 0.238 | 0.346 | 0.257 | 0.330 | 0.245 | 0.334 | 0.247 | 0.330 | 0.252 | 0.344 | 0.285 | 0.340 | 0.251 | 0.374 | 0.280 | 0.318 | 0.250 | 0.372 | 0.279 | 0.345 | 0.262 | 0.363 | 0.271 | 0.814 | 0.417 |
| | 12 | 0.310 | 0.230 | 0.303 | 0.224 | 0.311 | 0.232 | 0.302 | 0.227 | 0.325 | 0.250 | 0.326 | 0.236 | 0.330 | 0.255 | 0.326 | 0.252 | 0.379 | 0.285 | 0.364 | 0.281 | 0.370 | 0.282 | 0.303 | 0.238 | 0.369 | 0.268 | 0.345 | 0.253 | 0.353 | 0.270 | 0.801 | 0.403 |
| PEMS03 | 96 | 0.171 | 0.280 | 0.153 | 0.263 | 0.149 | 0.258 | 0.143 | 0.252 | 0.360 | 0.416 | 0.147 | 0.257 | 0.173 | 0.281 | 0.169 | 0.279 | 0.178 | 0.285 | 0.174 | 0.280 | 0.193 | 0.274 | 0.135 | 0.229 | 0.157 | 0.267 | 0.198 | 0.285 | 0.198 | 0.299 | 0.162 | 0.272 |
| | 48 | 0.122 | 0.234 | 0.106 | 0.217 | 0.105 | 0.215 | 0.102 | 0.210 | 0.299 | 0.380 | 0.109 | 0.216 | 0.128 | 0.235 | 0.123 | 0.248 | 0.131 | 0.243 | 0.126 | 0.237 | 0.155 | 0.247 | 0.108 | 0.215 | 0.116 | 0.230 | 0.167 | 0.266 | 0.155 | 0.260 | 0.117 | 0.234 |
| | 24 | 0.089 | 0.201 | 0.079 | 0.187 | 0.081 | 0.188 | 0.075 | 0.181 | 0.097 | 0.209 | 0.082 | 0.189 | 0.091 | 0.199 | 0.089 | 0.201 | 0.099 | 0.213 | 0.095 | 0.209 | 0.108 | 0.211 | 0.078 | 0.184 | 0.085 | 0.196 | 0.108 | 0.221 | 0.109 | 0.218 | 0.089 | 0.201 |
| | 12 | 0.067 | 0.173 | 0.063 | 0.167 | 0.065 | 0.169 | 0.062 | 0.165 | 0.078 | 0.185 | 0.067 | 0.177 | 0.071 | 0.178 | 0.068 | 0.176 | 0.077 | 0.187 | 0.070 | 0.176 | 0.076 | 0.180 | 0.062 | 0.166 | 0.061 | 0.166 | 0.088 | 0.209 | 0.075 | 0.184 | 0.075 | 0.186 |
| PEMS04 | 96 | 0.154 | 0.265 | 0.139 | 0.250 | 0.133 | 0.240 | 0.131 | 0.237 | 0.166 | 0.277 | 0.135 | 0.243 | 0.163 | 0.275 | 0.135 | 0.270 | 0.171 | 0.284 | 0.135 | 0.280 | 0.215 | 0.299 | 0.121 | 0.217 | 0.141 | 0.253 | 0.132 | 0.320 | 0.209 | 0.301 | 0.145 | 0.258 |
| | 48 | 0.133 | 0.246 | 0.119 | 0.230 | 0.114 | 0.223 | 0.106 | 0.217 | 0.299 | 0.389 | 0.094 | 0.225 | 0.138 | 0.251 | 0.103 | 0.247 | 0.138 | 0.253 | 0.107 | 0.250 | 0.173 | 0.270 | 0.108 | 0.205 | 0.126 | 0.247 | 0.214 | 0.340 | 0.167 | 0.268 | 0.104 | 0.216 |
| | 24 | 0.103 | 0.215 | 0.092 | 0.199 | 0.091 | 0.199 | 0.085 | 0.194 | 0.116 | 0.230 | 0.077 | 0.202 | 0.103 | 0.217 | 0.089 | 0.215 | 0.113 | 0.227 | 0.092 | 0.218 | 0.124 | 0.230 | 0.087 | 0.186 | 0.090 | 0.201 | 0.131 | 0.243 | 0.123 | 0.229 | 0.092 | 0.203 |
| | 12 | 0.082 | 0.190 | 0.075 | 0.175 | 0.075 | 0.180 | 0.070 | 0.176 | 0.087 | 0.195 | 0.078 | 0.183 | 0.085 | 0.191 | 0.068 | 0.188 | 0.089 | 0.198 | 0.073 | 0.193 | 0.092 | 0.199 | 0.073 | 0.174 | 0.074 | 0.180 | 0.066 | 0.196 | 0.091 | 0.197 | 0.085 | 0.193 |
| PEMS07 | 96 | 0.126 | 0.237 | 0.118 | 0.227 | 0.113 | 0.220 | 0.117 | 0.225 | 0.126 | 0.137 | 0.118 | 0.228 | 0.138 | 0.248 | 0.133 | 0.245 | 0.141 | 0.256 | 0.139 | 0.251 | 0.193 | 0.289 | 0.119 | 0.210 | 0.144 | 0.257 | 0.132 | 0.245 | 0.178 | 0.278 | 0.145 | 0.258 |
| | 48 | 0.091 | 0.201 | 0.082 | 0.189 | 0.084 | 0.185 | 0.106 | 0.179 | 0.093 | 0.204 | 0.085 | 0.199 | 0.109 | 0.222 | 0.106 | 0.219 | 0.114 | 0.227 | 0.109 | 0.224 | 0.152 | 0.254 | 0.072 | 0.171 | 0.102 | 0.212 | 0.091 | 0.206 | 0.155 | 0.257 | 0.104 | 0.216 |
| | 24 | 0.080 | 0.189 | 0.063 | 0.171 | 0.079 | 0.189 | 0.065 | 0.169 | 0.080 | 0.188 | 0.077 | 0.186 | 0.086 | 0.199 | 0.081 | 0.190 | 0.094 | 0.206 | 0.086 | 0.196 | 0.106 | 0.213 | 0.068 | 0.156 | 0.075 | 0.181 | 0.082 | 0.193 | 0.107 | 0.218 | 0.086 | 0.196 |
| | 12 | 0.072 | 0.180 | 0.063 | 0.168 | 0.072 | 0.180 | 0.055 | 0.152 | 0.072 | 0.184 | 0.068 | 0.180 | 0.071 | 0.181 | 0.069 | 0.178 | 0.073 | 0.184 | 0.067 | 0.174 | 0.075 | 0.178 | 0.055 | 0.150 | 0.056 | 0.154 | 0.066 | 0.162 | 0.074 | 0.179 | 0.073 | 0.180 |
| PEMS08 | 96 | 0.222 | 0.262 | 0.207 | 0.249 | 0.179 | 0.235 | 0.175 | 0.233 | 0.255 | 0.291 | 0.194 | 0.238 | 0.229 | 0.271 | 0.221 | 0.268 | 0.234 | 0.273 | 0.226 | 0.269 | 0.337 | 0.322 | 0.170 | 0.215 | 0.212 | 0.276 | 0.236 | 0.236 | 0.318 | 0.326 | 0.253 | 0.302 |
| | 48 | 0.166 | 0.241 | 0.141 | 0.222 | 0.128 | 0.215 | 0.125 | 0.208 | 0.191 | 0.269 | 0.138 | 0.228 | 0.172 | 0.248 | 0.164 | 0.247 | 0.188 | 0.263 | 0.164 | 0.248 | 0.232 | 0.282 | 0.131 | 0.197 | 0.142 | 0.238 | 0.136 | 0.225 | 0.223 | 0.283 | 0.203 | 0.279 |
| | 24 | 0.114 | 0.209 | 0.102 | 0.192 | 0.095 | 0.191 | 0.086 | 0.184 | 0.112 | 0.212 | 0.098 | 0.199 | 0.112 | 0.213 | 0.109 | 0.211 | 0.118 | 0.225 | 0.114 | 0.214 | 0.142 | 0.233 | 0.090 | 0.183 | 0.107 | 0.203 | 0.091 | 0.199 | 0.137 | 0.232 | 0.154 | 0.238 |
| | 12 | 0.088 | 0.189 | 0.077 | 0.173 | 0.074 | 0.171 | 0.076 | 0.175 | 0.079 | 0.182 | 0.071 | 0.170 | 0.086 | 0.185 | 0.082 | 0.186 | 0.103 | 0.225 | 0.105 | 0.221 | 0.094 | 0.196 | 0.076 | 0.171 | 0.078 | 0.177 | 0.073 | 0.176 | 0.092 | 0.195 | 0.123 | 0.216 |

Table 9: Results showing that ARMA Transformers with $m = 3$ layers consistently outperform their AR counterparts across a wide range of $m$. Forecasting horizons $L_P \in \{12, 24, 48, 96\}$ and $L_I = 512$ are used. Test set MSE and MAE for each model on each experiment setup are presented.

| | | Model | ARMA(m=3) | | AR(m=1) | | AR(m=2) | | AR(m=3) | | AR(m=4) | | AR(m=5) | | AR(m=6) | | AR(m=7) | | AR(m=8) | |
|---|---|---|---|---|---|---|---|---|---|---|---|---|---|---|---|---|---|---|---|---|
| | | Metrics | MSE | MAE | MSE | MAE | MSE | MAE | MSE | MAE | MSE | MAE | MSE | MAE | MSE | MAE | MSE | MAE | MSE | MAE |
| ETTm1 | 96 | Std Attn | 0.301 | 0.354 | 0.305 | 0.360 | 0.308 | 0.360 | 0.305 | 0.359 | 0.308 | 0.360 | 0.304 | 0.358 | 0.309 | 0.361 | 0.306 | 0.359 | 0.307 | 0.359 |
| | | Lin Attn | 0.296 | 0.351 | 0.310 | 0.361 | 0.301 | 0.358 | 0.303 | 0.355 | 0.303 | 0.356 | 0.299 | 0.352 | 0.301 | 0.355 | 0.299 | 0.353 | 0.300 | 0.354 |
| | | GLin Attn | 0.299 | 0.355 | 0.337 | 0.381 | 0.337 | 0.387 | 0.336 | 0.382 | 0.334 | 0.380 | 0.337 | 0.382 | 0.337 | 0.382 | 0.335 | 0.381 | 0.333 | 0.379 |
| | | ELin Attn | 0.301 | 0.354 | 0.307 | 0.363 | 0.309 | 0.361 | 0.305 | 0.356 | 0.307 | 0.360 | 0.306 | 0.359 | 0.309 | 0.362 | 0.305 | 0.359 | 0.308 | 0.361 |
| | | Fixed Attn | 0.296 | 0.344 | 0.299 | 0.346 | 0.298 | 0.349 | 0.298 | 0.347 | 0.299 | 0.347 | 0.300 | 0.348 | 0.298 | 0.347 | 0.302 | 0.348 | 0.305 | 0.351 |
| | 48 | Std Attn | 0.276 | 0.333 | 0.293 | 0.347 | 0.293 | 0.347 | 0.287 | 0.344 | 0.290 | 0.345 | 0.290 | 0.345 | 0.288 | 0.344 | 0.286 | 0.342 | 0.291 | 0.345 |
| | | Lin Attn | 0.266 | 0.328 | 0.280 | 0.336 | 0.278 | 0.337 | 0.278 | 0.336 | 0.278 | 0.336 | 0.271 | 0.331 | 0.272 | 0.333 | 0.274 | 0.332 | 0.276 | 0.334 |
| | | GLin Attn | 0.372 | 0.334 | 0.494 | 0.347 | 0.496 | 0.464 | 0.500 | 0.472 | 0.505 | 0.467 | 0.500 | 0.468 | 0.516 | 0.459 | 0.494 | 0.469 | 0.499 | 0.469 |
| | | ELin Attn | 0.282 | 0.342 | 0.293 | 0.350 | 0.289 | 0.350 | 0.284 | 0.346 | 0.292 | 0.349 | 0.294 | 0.352 | 0.295 | 0.352 | 0.292 | 0.350 | 0.299 | 0.355 |
| | | Fixed Attn | 0.280 | 0.340 | 0.286 | 0.345 | 0.283 | 0.340 | 0.284 | 0.338 | 0.283 | 0.340 | 0.284 | 0.338 | 0.277 | 0.335 | 0.282 | 0.335 | 0.279 | 0.336 |
| | 24 | Std Attn | 0.223 | 0.293 | 0.234 | 0.300 | 0.258 | 0.323 | 0.246 | 0.312 | 0.244 | 0.311 | 0.262 | 0.325 | 0.260 | 0.324 | 0.259 | 0.323 | 0.263 | 0.327 |
| | | Lin Attn | 0.196 | 0.279 | 0.226 | 0.297 | 0.210 | 0.288 | 0.218 | 0.293 | 0.211 | 0.288 | 0.210 | 0.286 | 0.208 | 0.285 | 0.212 | 0.287 | 0.209 | 0.287 |
| | | GLin Attn | 0.225 | 0.305 | 0.487 | 0.430 | 0.499 | 0.440 | 0.473 | 0.429 | 0.476 | 0.436 | 0.482 | 0.436 | 0.466 | 0.435 | 0.486 | 0.440 | 0.463 | 0.430 |
| | | ELin Attn | 0.241 | 0.307 | 0.253 | 0.328 | 0.246 | 0.317 | 0.239 | 0.305 | 0.253 | 0.318 | 0.268 | 0.327 | 0.263 | 0.322 | 0.264 | 0.325 | 0.264 | 0.326 |
| | | Fixed Attn | 0.255 | 0.321 | 0.274 | 0.317 | 0.272 | 0.334 | 0.284 | 0.332 | 0.260 | 0.326 | 0.254 | 0.320 | 0.266 | 0.329 | 0.257 | 0.322 | 0.255 | 0.322 |
| | 12 | Std Attn | 0.156 | 0.241 | 0.229 | 0.293 | 0.222 | 0.288 | 0.218 | 0.287 | 0.221 | 0.291 | 0.221 | 0.289 | 0.223 | 0.291 | 0.228 | 0.288 | 0.227 | 0.287 |
| | | Lin Attn | 0.128 | 0.222 | 0.148 | 0.239 | 0.141 | 0.232 | 0.151 | 0.240 | 0.137 | 0.228 | 0.138 | 0.234 | 0.139 | 0.231 | 0.137 | 0.228 | 0.138 | 0.229 |
| | | GLin Attn | 0.144 | 0.237 | 0.325 | 0.325 | 0.321 | 0.324 | 0.320 | 0.335 | 0.319 | 0.326 | 0.322 | 0.326 | 0.322 | 0.325 | 0.320 | 0.323 | 0.320 | 0.323 |
| | | ELin Attn | 0.153 | 0.246 | 0.160 | 0.251 | 0.160 | 0.252 | 0.157 | 0.247 | 0.159 | 0.250 | 0.160 | 0.252 | 0.169 | 0.260 | 0.163 | 0.251 | 0.160 | 0.247 |
| | | Fixed Attn | 0.174 | 0.261 | 0.215 | 0.289 | 0.204 | 0.285 | 0.203 | 0.282 | 0.199 | 0.281 | 0.194 | 0.279 | 0.195 | 0.278 | 0.192 | 0.275 | 0.189 | 0.278 |
| Weather | 96 | Std Attn | 0.142 | 0.193 | 0.156 | 0.207 | 0.153 | 0.210 | 0.144 | 0.195 | 0.152 | 0.206 | 0.156 | 0.201 | 0.156 | 0.209 | 0.156 | 0.207 | 0.156 | 0.207 |
| | | Lin Attn | 0.139 | 0.191 | 0.144 | 0.197 | 0.143 | 0.195 | 0.142 | 0.194 | 0.143 | 0.196 | 0.143 | 0.194 | 0.143 | 0.194 | 0.142 | 0.193 | 0.144 | 0.196 |
| | | GLin Attn | 0.142 | 0.194 | 0.163 | 0.213 | 0.163 | 0.212 | 0.161 | 0.210 | 0.165 | 0.213 | 0.163 | 0.212 | 0.164 | 0.213 | 0.164 | 0.216 | 0.164 | 0.213 |
| | | ELin Attn | 0.143 | 0.195 | 0.157 | 0.211 | 0.148 | 0.207 | 0.146 | 0.197 | 0.151 | 0.211 | 0.156 | 0.208 | 0.157 | 0.211 | 0.156 | 0.207 | 0.157 | 0.208 |
| | | Fixed Attn | 0.142 | 0.198 | 0.158 | 0.210 | 0.151 | 0.209 | 0.147 | 0.194 | 0.152 | 0.206 | 0.154 | 0.206 | 0.153 | 0.204 | 0.153 | 0.207 | 0.153 | 0.205 |
| | 48 | Std Attn | 0.109 | 0.151 | 0.116 | 0.161 | 0.115 | 0.159 | 0.113 | 0.157 | 0.116 | 0.160 | 0.127 | 0.177 | 0.120 | 0.168 | 0.128 | 0.181 | 0.127 | 0.177 |
| | | Lin Attn | 0.110 | 0.153 | 0.114 | 0.156 | 0.115 | 0.159 | 0.115 | 0.158 | 0.113 | 0.153 | 0.113 | 0.155 | 0.112 | 0.153 | 0.112 | 0.155 | 0.112 | 0.155 |
| | | GLin Attn | 0.116 | 0.159 | 0.144 | 0.190 | 0.144 | 0.189 | 0.144 | 0.191 | 0.145 | 0.193 | 0.145 | 0.191 | 0.144 | 0.192 | 0.143 | 0.189 | 0.143 | 0.189 |
| | | ELin Attn | 0.112 | 0.156 | 0.119 | 0.167 | 0.116 | 0.166 | 0.114 | 0.158 | 0.117 | 0.166 | 0.118 | 0.165 | 0.122 | 0.172 | 0.121 | 0.167 | 0.123 | 0.171 |
| | | Fixed Attn | 0.115 | 0.159 | 0.124 | 0.171 | 0.118 | 0.170 | 0.112 | 0.155 | 0.122 | 0.170 | 0.126 | 0.173 | 0.121 | 0.171 | 0.122 | 0.168 | 0.121 | 0.167 |
| | 24 | Std Attn | 0.085 | 0.115 | 0.091 | 0.124 | 0.091 | 0.124 | 0.089 | 0.118 | 0.093 | 0.124 | 0.096 | 0.132 | 0.095 | 0.130 | 0.095 | 0.129 | 0.094 | 0.126 |
| | | Lin Attn | 0.083 | 0.114 | 0.087 | 0.117 | 0.087 | 0.118 | 0.088 | 0.117 | 0.087 | 0.115 | 0.086 | 0.116 | 0.087 | 0.118 | 0.087 | 0.117 | 0.087 | 0.116 |
| | | GLin Attn | 0.090 | 0.122 | 0.103 | 0.132 | 0.103 | 0.132 | 0.101 | 0.129 | 0.103 | 0.132 | 0.103 | 0.129 | 0.103 | 0.134 | 0.104 | 0.132 | 0.103 | 0.132 |
| | | ELin Attn | 0.089 | 0.119 | 0.092 | 0.123 | 0.092 | 0.123 | 0.087 | 0.116 | 0.090 | 0.122 | 0.092 | 0.128 | 0.092 | 0.128 | 0.092 | 0.127 | 0.092 | 0.124 |
| | | Fixed Attn | 0.088 | 0.120 | 0.095 | 0.127 | 0.093 | 0.126 | 0.091 | 0.122 | 0.094 | 0.128 | 0.094 | 0.125 | 0.093 | 0.125 | 0.092 | 0.124 | 0.094 | 0.127 |
| | 12 | Std Attn | 0.067 | 0.086 | 0.073 | 0.091 | 0.072 | 0.091 | 0.071 | 0.091 | 0.072 | 0.093 | 0.072 | 0.091 | 0.072 | 0.091 | 0.072 | 0.094 | 0.072 | 0.091 |
| | | Lin Attn | 0.067 | 0.086 | 0.069 | 0.088 | 0.069 | 0.089 | 0.069 | 0.088 | 0.069 | 0.089 | 0.068 | 0.088 | 0.068 | 0.087 | 0.068 | 0.088 | 0.069 | 0.090 |
| | | GLin Attn | 0.070 | 0.091 | 0.078 | 0.092 | 0.077 | 0.095 | 0.070 | 0.090 | 0.072 | 0.093 | 0.071 | 0.088 | 0.077 | 0.093 | 0.071 | 0.088 | 0.071 | 0.090 |
| | | ELin Attn | 0.069 | 0.087 | 0.071 | 0.090 | 0.071 | 0.091 | 0.069 | 0.087 | 0.072 | 0.091 | 0.071 | 0.091 | 0.071 | 0.090 | 0.072 | 0.091 | 0.071 | 0.090 |
| | | Fixed Attn | 0.069 | 0.090 | 0.073 | 0.093 | 0.072 | 0.094 | 0.071 | 0.089 | 0.072 | 0.093 | 0.072 | 0.091 | 0.072 | 0.090 | 0.072 | 0.089 | 0.072 | 0.091 |

Table 10: Results showing that AR/ARMA Transformers effectively utilize extended lookback $L_I$, while baselines experience performance degradation. $L_I \in \{512, 1024, 2048, 4096\}$ with $L_P \in \{12, 24, 48, 96\}$ are evaluated. Test set MSE and MAE for each model on each setup are presented.

| Dataset | $L_I$ | $L_P$ | Std Attn | | Std Attn +ARMA | | Lin Attn | | Lin Attn +ARMA | | GLin Attn | | GLin Attn +ARMA | | ELin Attn | | ELin Attn +ARMA | | Fixed Attn | | Fixed Attn +ARMA | | FITS | | iTransformer | | CATS | | PatchTST | | DLinear | | EncFormer | |
|---|---|---|---|---|---|---|---|---|---|---|---|---|---|---|---|---|---|---|---|---|---|---|---|---|---|---|---|---|---|---|---|---|---|---|---|---|
| | | | MSE | MAE | MSE | MAE | MSE | MAE | MSE | MAE | MSE | MAE | MSE | MAE | MSE | MAE | MSE | MAE | MSE | MAE | MSE | MAE | MSE | MAE | MSE | MAE | MSE | MAE | MSE | MAE | MSE | MAE | MSE | MAE |
| Weather | 512 | 96 | 0.144 | 0.195 | 0.142 | 0.193 | 0.142 | 0.194 | 0.139 | 0.191 | 0.161 | 0.210 | 0.142 | 0.194 | 0.145 | 0.197 | 0.143 | 0.195 | 0.147 | 0.194 | 0.142 | 0.198 | 0.151 | 0.204 | 0.158 | 0.210 | 0.146 | 0.198 | 0.151 | 0.224 | 0.150 | 0.209 | 0.188 | 0.248 |
| | 512 | 48 | 0.113 | 0.157 | 0.109 | 0.151 | 0.115 | 0.158 | 0.110 | 0.153 | 0.144 | 0.191 | 0.116 | 0.159 | 0.114 | 0.159 | 0.112 | 0.156 | 0.112 | 0.155 | 0.115 | 0.159 | 0.125 | 0.177 | 0.132 | 0.176 | 0.116 | 0.160 | 0.118 | 0.161 | 0.122 | 0.177 | 0.143 | 0.199 |
| | 512 | 24 | 0.089 | 0.118 | 0.085 | 0.115 | 0.088 | 0.117 | 0.083 | 0.114 | 0.101 | 0.129 | 0.090 | 0.122 | 0.087 | 0.116 | 0.089 | 0.119 | 0.091 | 0.122 | 0.088 | 0.120 | 0.103 | 0.147 | 0.101 | 0.139 | 0.090 | 0.125 | 0.092 | 0.122 | 0.147 | 0.102 | 0.117 | 0.162 |
| | 512 | 12 | 0.071 | 0.091 | 0.067 | 0.086 | 0.069 | 0.088 | 0.067 | 0.086 | 0.070 | 0.090 | 0.070 | 0.091 | 0.069 | 0.087 | 0.069 | 0.087 | 0.071 | 0.089 | 0.069 | 0.090 | 0.078 | 0.112 | 0.078 | 0.105 | 0.069 | 0.092 | 0.071 | 0.097 | 0.078 | 0.115 | 0.093 | 0.120 |
| | 1024 | 96 | 0.145 | 0.196 | 0.142 | 0.194 | 0.144 | 0.198 | 0.141 | 0.195 | 0.161 | 0.212 | 0.145 | 0.198 | 0.147 | 0.198 | 0.144 | 0.197 | 0.149 | 0.201 | 0.144 | 0.198 | 0.168 | 0.222 | 0.164 | 0.219 | 0.148 | 0.203 | 0.167 | 0.223 | 0.166 | 0.222 | 0.175 | 0.247 |
| | 1024 | 48 | 0.112 | 0.156 | 0.110 | 0.152 | 0.110 | 0.154 | 0.109 | 0.152 | 0.144 | 0.193 | 0.115 | 0.157 | 0.114 | 0.159 | 0.117 | 0.156 | 0.116 | 0.161 | 0.114 | 0.162 | 0.132 | 0.185 | 0.127 | 0.181 | 0.118 | 0.164 | 0.133 | 0.187 | 0.129 | 0.181 | 0.141 | 0.202 |
| | 1024 | 24 | 0.096 | 0.130 | 0.088 | 0.118 | 0.086 | 0.112 | 0.086 | 0.116 | 0.091 | 0.123 | 0.089 | 0.122 | 0.090 | 0.122 | 0.091 | 0.123 | 0.095 | 0.129 | 0.092 | 0.126 | 0.102 | 0.147 | 0.098 | 0.141 | 0.091 | 0.128 | 0.101 | 0.144 | 0.100 | 0.144 | 0.104 | 0.146 |
| | 1024 | 12 | 0.075 | 0.092 | 0.068 | 0.086 | 0.068 | 0.088 | 0.067 | 0.087 | 0.069 | 0.090 | 0.068 | 0.087 | 0.073 | 0.092 | 0.071 | 0.090 | 0.072 | 0.092 | 0.071 | 0.091 | 0.078 | 0.112 | 0.078 | 0.114 | 0.073 | 0.102 | 0.077 | 0.110 | 0.077 | 0.112 | 0.075 | 0.119 |
| | 2048 | 96 | 0.154 | 0.208 | 0.144 | 0.200 | 0.139 | 0.195 | 0.139 | 0.195 | 0.145 | 0.216 | 0.144 | 0.200 | 0.155 | 0.210 | 0.153 | 0.209 | 0.155 | 0.213 | 0.152 | 0.216 | 0.169 | 0.225 | 0.164 | 0.220 | 0.154 | 0.212 | 0.168 | 0.224 | 0.167 | 0.223 | 0.194 | 0.270 |
| | 2048 | 48 | 0.114 | 0.159 | 0.110 | 0.157 | 0.110 | 0.155 | 0.108 | 0.152 | 0.146 | 0.198 | 0.110 | 0.155 | 0.115 | 0.163 | 0.115 | 0.163 | 0.132 | 0.184 | 0.123 | 0.173 | 0.134 | 0.187 | 0.128 | 0.184 | 0.122 | 0.176 | 0.135 | 0.190 | 0.131 | 0.185 | 0.129 | 0.190 |
| | 2048 | 24 | 0.097 | 0.135 | 0.085 | 0.119 | 0.086 | 0.115 | 0.085 | 0.117 | 0.096 | 0.122 | 0.087 | 0.118 | 0.090 | 0.125 | 0.090 | 0.125 | 0.096 | 0.125 | 0.093 | 0.128 | 0.102 | 0.149 | 0.101 | 0.150 | 0.096 | 0.135 | 0.103 | 0.152 | 0.100 | 0.146 | 0.108 | 0.157 |
| | 2048 | 12 | 0.073 | 0.094 | 0.068 | 0.089 | 0.068 | 0.088 | 0.067 | 0.086 | 0.069 | 0.090 | 0.068 | 0.089 | 0.072 | 0.093 | 0.072 | 0.093 | 0.073 | 0.095 | 0.072 | 0.093 | 0.080 | 0.120 | 0.082 | 0.127 | 0.078 | 0.104 | 0.080 | 0.124 | 0.077 | 0.117 | 0.079 | 0.111 |
| | 4096 | 96 | 0.150 | 0.207 | 0.143 | 0.203 | 0.139 | 0.198 | 0.139 | 0.201 | 0.166 | 0.220 | 0.142 | 0.201 | 0.153 | 0.213 | 0.150 | 0.212 | 0.157 | 0.215 | 0.155 | 0.213 | 0.168 | 0.228 | 0.168 | 0.231 | 0.164 | 0.225 | 0.171 | 0.232 | 0.171 | 0.235 | 0.212 | 0.281 |
| | 4096 | 48 | 0.114 | 0.162 | 0.111 | 0.159 | 0.109 | 0.155 | 0.107 | 0.156 | 0.137 | 0.193 | 0.123 | 0.176 | 0.116 | 0.168 | 0.116 | 0.166 | 0.117 | 0.168 | 0.111 | 0.159 | 0.135 | 0.194 | 0.173 | 0.201 | 0.138 | 0.194 | 0.138 | 0.199 | 0.132 | 0.191 | 0.139 | 0.194 |
| | 4096 | 24 | 0.097 | 0.140 | 0.085 | 0.122 | 0.085 | 0.117 | 0.085 | 0.118 | 0.086 | 0.121 | 0.086 | 0.120 | 0.093 | 0.130 | 0.089 | 0.125 | 0.096 | 0.132 | 0.094 | 0.129 | 0.105 | 0.156 | 0.103 | 0.154 | 0.108 | 0.153 | 0.107 | 0.162 | 0.102 | 0.152 | 0.114 | 0.160 |
| | 4096 | 12 | 0.072 | 0.096 | 0.067 | 0.089 | 0.068 | 0.088 | 0.068 | 0.088 | 0.070 | 0.092 | 0.068 | 0.089 | 0.071 | 0.096 | 0.071 | 0.097 | 0.071 | 0.097 | 0.071 | 0.096 | 0.086 | 0.135 | 0.085 | 0.133 | 0.081 | 0.118 | 0.085 | 0.140 | 0.079 | 0.117 | 0.080 | 0.110 |
| ETTm1 | 512 | 96 | 0.305 | 0.359 | 0.301 | 0.354 | 0.303 | 0.355 | 0.296 | 0.351 | 0.336 | 0.382 | 0.299 | 0.355 | 0.305 | 0.356 | 0.301 | 0.354 | 0.298 | 0.347 | 0.296 | 0.344 | 0.305 | 0.347 | 0.352 | 0.378 | 0.300 | 0.354 | 0.323 | 0.362 | 0.305 | 0.348 | 0.686 | 0.603 |
| | 512 | 48 | 0.287 | 0.344 | 0.276 | 0.333 | 0.278 | 0.336 | 0.266 | 0.328 | 0.500 | 0.472 | 0.372 | 0.334 | 0.284 | 0.346 | 0.282 | 0.342 | 0.284 | 0.338 | 0.280 | 0.340 | 0.280 | 0.331 | 0.304 | 0.346 | 0.266 | 0.328 | 0.289 | 0.341 | 0.278 | 0.329 | 0.475 | 0.474 |
| | 512 | 24 | 0.246 | 0.312 | 0.223 | 0.293 | 0.196 | 0.293 | 0.196 | 0.279 | 0.473 | 0.429 | 0.225 | 0.305 | 0.239 | 0.305 | 0.241 | 0.307 | 0.284 | 0.332 | 0.255 | 0.321 | 0.218 | 0.289 | 0.223 | 0.293 | 0.197 | 0.277 | 0.235 | 0.304 | 0.216 | 0.288 | 0.352 | 0.391 |
| | 512 | 12 | 0.218 | 0.287 | 0.156 | 0.241 | 0.151 | 0.240 | 0.128 | 0.222 | 0.320 | 0.335 | 0.144 | 0.237 | 0.157 | 0.247 | 0.153 | 0.246 | 0.203 | 0.282 | 0.174 | 0.261 | 0.144 | 0.235 | 0.155 | 0.246 | 0.125 | 0.222 | 0.128 | 0.224 | 0.140 | 0.230 | 0.204 | 0.294 |
| | 1024 | 96 | 0.318 | 0.366 | 0.308 | 0.357 | 0.308 | 0.356 | 0.304 | 0.357 | 0.339 | 0.384 | 0.339 | 0.362 | 0.317 | 0.367 | 0.308 | 0.359 | 0.306 | 0.356 | 0.303 | 0.353 | 0.310 | 0.353 | 0.340 | 0.377 | 0.314 | 0.367 | 0.321 | 0.363 | 0.308 | 0.353 | 0.493 | 0.497 |
| | 1024 | 48 | 0.308 | 0.355 | 0.281 | 0.333 | 0.292 | 0.345 | 0.272 | 0.333 | 0.532 | 0.486 | 0.277 | 0.340 | 0.315 | 0.363 | 0.296 | 0.349 | 0.303 | 0.352 | 0.300 | 0.351 | 0.285 | 0.337 | 0.310 | 0.350 | 0.293 | 0.346 | 0.290 | 0.340 | 0.287 | 0.344 | 0.442 | 0.423 |
| | 1024 | 24 | 0.267 | 0.332 | 0.231 | 0.305 | 0.202 | 0.294 | 0.202 | 0.282 | 0.504 | 0.441 | 0.218 | 0.298 | 0.266 | 0.328 | 0.252 | 0.328 | 0.292 | 0.352 | 0.271 | 0.330 | 0.222 | 0.293 | 0.228 | 0.301 | 0.210 | 0.289 | 0.219 | 0.292 | 0.219 | 0.292 | 0.338 | 0.373 |
| | 1024 | 12 | 0.228 | 0.236 | 0.145 | 0.299 | 0.138 | 0.229 | 0.130 | 0.222 | 0.317 | 0.323 | 0.140 | 0.232 | 0.160 | 0.248 | 0.157 | 0.251 | 0.221 | 0.292 | 0.179 | 0.269 | 0.144 | 0.237 | 0.153 | 0.249 | 0.133 | 0.229 | 0.151 | 0.245 | 0.140 | 0.232 | 0.182 | 0.268 |
| | 2048 | 96 | 0.314 | 0.364 | 0.302 | 0.357 | 0.301 | 0.356 | 0.301 | 0.356 | 0.342 | 0.387 | 0.304 | 0.360 | 0.313 | 0.366 | 0.305 | 0.360 | 0.304 | 0.358 | 0.304 | 0.356 | 0.311 | 0.356 | 0.330 | 0.373 | 0.335 | 0.384 | 0.318 | 0.366 | 0.311 | 0.360 | 0.503 | 0.504 |
| | 2048 | 48 | 0.302 | 0.353 | 0.281 | 0.334 | 0.282 | 0.339 | 0.266 | 0.331 | 0.529 | 0.482 | 0.272 | 0.338 | 0.298 | 0.364 | 0.298 | 0.355 | 0.312 | 0.369 | 0.314 | 0.369 | 0.287 | 0.341 | 0.290 | 0.341 | 0.307 | 0.359 | 0.305 | 0.361 | 0.281 | 0.339 | 0.529 | 0.563 |
| | 2048 | 24 | 0.270 | 0.337 | 0.228 | 0.305 | 0.214 | 0.290 | 0.198 | 0.283 | 0.272 | 0.330 | 0.211 | 0.293 | 0.273 | 0.330 | 0.244 | 0.314 | 0.312 | 0.367 | 0.279 | 0.338 | 0.221 | 0.298 | 0.225 | 0.304 | 0.253 | 0.324 | 0.217 | 0.294 | 0.220 | 0.298 | 0.409 | 0.436 |
| | 2048 | 12 | 0.226 | 0.238 | 0.146 | 0.296 | 0.136 | 0.232 | 0.136 | 0.222 | 0.164 | 0.255 | 0.140 | 0.238 | 0.162 | 0.295 | 0.162 | 0.252 | 0.225 | 0.294 | 0.173 | 0.260 | 0.163 | 0.258 | 0.147 | 0.201 | 0.148 | 0.251 | 0.160 | 0.257 | 0.142 | 0.239 | 0.218 | 0.306 |
| | 4096 | 96 | 0.305 | 0.365 | 0.296 | 0.356 | 0.299 | 0.361 | 0.298 | 0.359 | 0.338 | 0.385 | 0.296 | 0.356 | 0.306 | 0.366 | 0.298 | 0.359 | 0.303 | 0.362 | 0.299 | 0.356 | 0.319 | 0.366 | 0.357 | 0.394 | 0.454 | 0.455 | 0.317 | 0.364 | 0.324 | 0.368 | 0.619 | 0.576 |
| | 4096 | 48 | 0.298 | 0.354 | 0.276 | 0.339 | 0.292 | 0.350 | 0.274 | 0.342 | 0.551 | 0.502 | 0.272 | 0.341 | 0.318 | 0.374 | 0.293 | 0.357 | 0.332 | 0.386 | 0.316 | 0.370 | 0.299 | 0.353 | 0.315 | 0.365 | 0.387 | 0.415 | 0.321 | 0.373 | 0.296 | 0.354 | 0.481 | 0.509 |
| | 4096 | 24 | 0.266 | 0.331 | 0.227 | 0.302 | 0.220 | 0.303 | 0.203 | 0.291 | 0.252 | 0.322 | 0.211 | 0.298 | 0.269 | 0.333 | 0.252 | 0.325 | 0.343 | 0.386 | 0.279 | 0.344 | 0.231 | 0.308 | 0.245 | 0.323 | 0.306 | 0.351 | 0.225 | 0.300 | 0.230 | 0.309 | 0.368 | 0.406 |
| | 4096 | 12 | 0.230 | 0.298 | 0.138 | 0.233 | 0.135 | 0.231 | 0.129 | 0.224 | 0.156 | 0.246 | 0.137 | 0.232 | 0.233 | 0.299 | 0.217 | 0.292 | 0.170 | 0.298 | 0.168 | 0.260 | 0.160 | 0.260 | 0.178 | 0.274 | 0.213 | 0.307 | 0.176 | 0.278 | 0.148 | 0.251 | 0.243 | 0.359 |

Table 11: Results of model performance on varying horizons $L_P$. AR/ARMA Transformers uses $L_I = 512$ for $L_P = 1$ and $L_I = 4096$ for $L_P = 720$. Baselines are consistently set to their best-performing $L_I = 512$ configuration. Test set MSE and MAE for each model on each setup are reported.

| Model | | Std Attn | | Std Attn +ARMA | | Lin Attn | | Lin Attn +ARMA | | GLin Attn | | GLin Attn +ARMA | | ELin Attn | | ELin Attn +ARMA | | Fixed Attn | | Fixed Attn +ARMA | | FITS | | iTransformer | | CATS | | PatchTST | | DLinear | | EncFormer | |
|---|---|---|---|---|---|---|---|---|---|---|---|---|---|---|---|---|---|---|---|---|---|---|---|---|---|---|---|---|---|---|---|---|---|
| Metrics | | MSE | MAE | MSE | MAE | MSE | MAE | MSE | MAE | MSE | MAE | MSE | MAE | MSE | MAE | MSE | MAE | MSE | MAE | MSE | MAE | MSE | MAE | MSE | MAE | MSE | MAE | MSE | MAE | MSE | MAE | MSE | MAE |
| ETTm1 | 1 | 0.048 | 0.140 | 0.043 | 0.130 | 0.042 | 0.126 | 0.041 | 0.123 | 0.051 | 0.138 | 0.043 | 0.126 | 0.053 | 0.140 | 0.051 | 0.140 | 0.054 | 0.142 | 0.052 | 0.140 | 0.047 | 0.138 | 0.049 | 0.151 | 0.043 | 0.127 | 0.046 | 0.133 | 0.044 | 0.130 | 0.044 | 0.129 |
| | 720 | 0.396 | 0.418 | 0.391 | 0.416 | 0.417 | 0.435 | 0.408 | 0.429 | 0.396 | 0.419 | 0.393 | 0.418 | 0.395 | 0.419 | 0.391 | 0.416 | 0.403 | 0.423 | 0.399 | 0.421 | 0.420 | 0.412 | 0.438 | 0.440 | 0.418 | 0.415 | 0.408 | 0.422 | 0.420 | 0.416 | 0.802 | 0.676 |
| ETTm2 | 1 | 0.034 | 0.096 | 0.030 | 0.095 | 0.032 | 0.094 | 0.029 | 0.092 | 0.033 | 0.093 | 0.030 | 0.092 | 0.035 | 0.095 | 0.033 | 0.093 | 0.033 | 0.095 | 0.031 | 0.093 | 0.034 | 0.108 | 0.034 | 0.112 | 0.032 | 0.079 | 0.033 | 0.102 | 0.031 | 0.098 | 0.035 | 0.118 |
| | 720 | 0.329 | 0.373 | 0.327 | 0.371 | 0.341 | 0.383 | 0.338 | 0.381 | 0.334 | 0.373 | 0.328 | 0.374 | 0.327 | 0.372 | 0.326 | 0.371 | 0.331 | 0.378 | 0.327 | 0.373 | 0.360 | 0.381 | 0.369 | 0.391 | 0.351 | 0.379 | 0.362 | 0.383 | 0.403 | 0.426 | 2.641 | 1.387 |
| Weather | 1 | 0.037 | 0.046 | 0.032 | 0.044 | 0.032 | 0.042 | 0.031 | 0.041 | 0.033 | 0.044 | 0.031 | 0.041 | 0.037 | 0.049 | 0.034 | 0.047 | 0.036 | 0.045 | 0.034 | 0.044 | 0.038 | 0.059 | 0.038 | 0.064 | 0.033 | 0.046 | 0.035 | 0.054 | 0.034 | 0.052 | 0.032 | 0.046 |
| | 720 | 0.299 | 0.328 | 0.301 | 0.332 | 0.308 | 0.336 | 0.305 | 0.337 | 0.310 | 0.334 | 0.299 | 0.332 | 0.296 | 0.326 | 0.295 | 0.325 | 0.299 | 0.327 | 0.297 | 0.326 | 0.327 | 0.342 | 0.317 | 0.335 | 0.311 | 0.334 | 0.325 | 0.340 | 0.319 | 0.356 | 0.404 | 0.423 |

Table 12: Experiment results of the performance comparison with MEGA with $L_P \in \{12, 24, 48, 96\}$.

| Model | | Std Attn | | Std Attn +ARMA | | Lin Attn | | Lin Attn +ARMA | | GLin Attn | | GLin Attn +ARMA | | MEGA | |
|---|---|---|---|---|---|---|---|---|---|---|---|---|---|---|---|
| Metrics | | MSE | MAE | MSE | MAE | MSE | MAE | MSE | MAE | MSE | MAE | MSE | MAE | MSE | MAE |
| Weather | 96 | 0.144 | 0.195 | 0.142 | 0.193 | 0.142 | 0.194 | 0.139 | 0.191 | 0.161 | 0.210 | 0.142 | 0.194 | 0.164 | 0.212 |
| | 48 | 0.113 | 0.157 | 0.109 | 0.151 | 0.115 | 0.158 | 0.110 | 0.153 | 0.144 | 0.191 | 0.116 | 0.159 | 0.141 | 0.187 |
| | 24 | 0.089 | 0.118 | 0.085 | 0.115 | 0.088 | 0.117 | 0.083 | 0.114 | 0.101 | 0.129 | 0.090 | 0.122 | 0.102 | 0.128 |
| | 12 | 0.071 | 0.091 | 0.067 | 0.086 | 0.069 | 0.088 | 0.067 | 0.086 | 0.070 | 0.090 | 0.070 | 0.091 | 0.077 | 0.090 |
| Solar | 96 | 0.196 | 0.263 | 0.192 | 0.257 | 0.183 | 0.247 | 0.180 | 0.244 | 0.209 | 0.283 | 0.182 | 0.243 | 0.235 | 0.302 |
| | 48 | 0.160 | 0.230 | 0.151 | 0.223 | 0.152 | 0.219 | 0.149 | 0.217 | 0.177 | 0.258 | 0.154 | 0.222 | 0.342 | 0.406 |
| | 24 | 0.112 | 0.180 | 0.098 | 0.168 | 0.098 | 0.166 | 0.095 | 0.162 | 0.143 | 0.232 | 0.099 | 0.167 | 0.231 | 0.284 |
| | 12 | 0.069 | 0.137 | 0.055 | 0.118 | 0.056 | 0.113 | 0.052 | 0.111 | 0.063 | 0.139 | 0.059 | 0.121 | 0.097 | 0.170 |
| ETTh1 | 96 | 0.357 | 0.393 | 0.360 | 0.395 | 0.358 | 0.396 | 0.361 | 0.399 | 0.378 | 0.406 | 0.368 | 0.404 | 0.373 | 0.468 |
| | 48 | 0.334 | 0.375 | 0.331 | 0.374 | 0.331 | 0.375 | 0.331 | 0.376 | 0.349 | 0.386 | 0.337 | 0.381 | 0.348 | 0.386 |
| | 24 | 0.312 | 0.365 | 0.299 | 0.357 | 0.299 | 0.356 | 0.299 | 0.357 | 0.349 | 0.394 | 0.303 | 0.360 | 0.345 | 0.387 |
| | 12 | 0.290 | 0.345 | 0.280 | 0.340 | 0.285 | 0.342 | 0.272 | 0.337 | 0.554 | 0.503 | 0.277 | 0.340 | 0.551 | 0.499 |
| ETTh2 | 96 | 0.266 | 0.330 | 0.268 | 0.331 | 0.273 | 0.336 | 0.275 | 0.338 | 0.285 | 0.338 | 0.281 | 0.345 | 0.278 | 0.330 |
| | 48 | 0.213 | 0.290 | 0.216 | 0.294 | 0.215 | 0.290 | 0.217 | 0.293 | 0.233 | 0.294 | 0.220 | 0.295 | 0.230 | 0.295 |
| | 24 | 0.160 | 0.252 | 0.160 | 0.252 | 0.159 | 0.250 | 0.162 | 0.252 | 0.182 | 0.263 | 0.164 | 0.254 | 0.180 | 0.261 |
| | 12 | 0.129 | 0.229 | 0.125 | 0.224 | 0.124 | 0.224 | 0.125 | 0.224 | 0.168 | 0.263 | 0.127 | 0.224 | 0.167 | 0.263 |
| ETTm1 | 96 | 0.305 | 0.359 | 0.301 | 0.354 | 0.303 | 0.355 | 0.296 | 0.351 | 0.336 | 0.382 | 0.299 | 0.355 | 0.335 | 0.378 |
| | 48 | 0.287 | 0.344 | 0.276 | 0.333 | 0.278 | 0.336 | 0.266 | 0.328 | 0.500 | 0.472 | 0.372 | 0.334 | 0.507 | 0.469 |
| | 24 | 0.246 | 0.312 | 0.223 | 0.293 | 0.218 | 0.293 | 0.196 | 0.279 | 0.473 | 0.429 | 0.225 | 0.305 | 0.487 | 0.434 |
| | 12 | 0.218 | 0.287 | 0.156 | 0.241 | 0.151 | 0.240 | 0.128 | 0.222 | 0.320 | 0.335 | 0.144 | 0.237 | 0.318 | 0.322 |
| ETTm2 | 96 | 0.177 | 0.262 | 0.174 | 0.261 | 0.167 | 0.255 | 0.162 | 0.250 | 0.178 | 0.260 | 0.172 | 0.258 | 0.176 | 0.258 |
| | 48 | 0.139 | 0.238 | 0.137 | 0.236 | 0.145 | 0.248 | 0.143 | 0.250 | 0.167 | 0.264 | 0.148 | 0.249 | 0.157 | 0.253 |
| | 24 | 0.121 | 0.219 | 0.117 | 0.217 | 0.110 | 0.211 | 0.101 | 0.199 | 0.132 | 0.227 | 0.110 | 0.209 | 0.126 | 0.223 |
| | 12 | 0.086 | 0.175 | 0.083 | 0.174 | 0.083 | 0.174 | 0.078 | 0.169 | 0.089 | 0.178 | 0.082 | 0.172 | 0.088 | 0.178 |
| PEMS03 | 96 | 0.171 | 0.280 | 0.153 | 0.263 | 0.149 | 0.258 | 0.143 | 0.252 | 0.360 | 0.416 | 0.147 | 0.257 | 0.223 | 0.329 |
| | 48 | 0.122 | 0.234 | 0.106 | 0.217 | 0.105 | 0.215 | 0.102 | 0.210 | 0.299 | 0.380 | 0.109 | 0.216 | 0.202 | 0.317 |
| | 24 | 0.089 | 0.201 | 0.079 | 0.187 | 0.081 | 0.188 | 0.075 | 0.181 | 0.097 | 0.209 | 0.082 | 0.189 | 0.129 | 0.244 |
| | 12 | 0.067 | 0.173 | 0.063 | 0.167 | 0.065 | 0.169 | 0.062 | 0.165 | 0.078 | 0.185 | 0.067 | 0.177 | 0.089 | 0.199 |

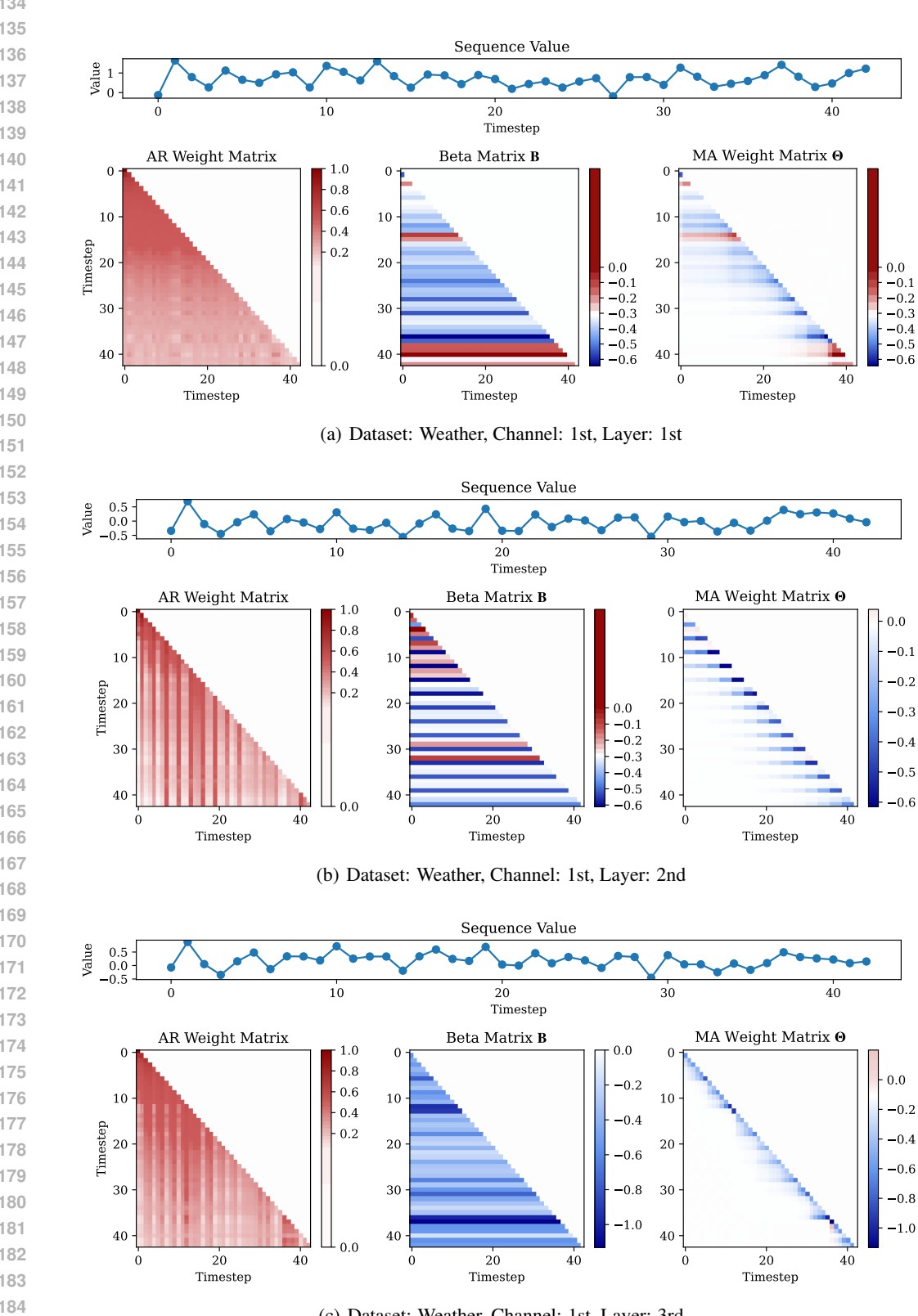

Figure 9: Visualization of the ARMA attention weights of the first input channel for the first test set data point in the Weather dataset ($L_I = 4096$, $L_P = 96$).

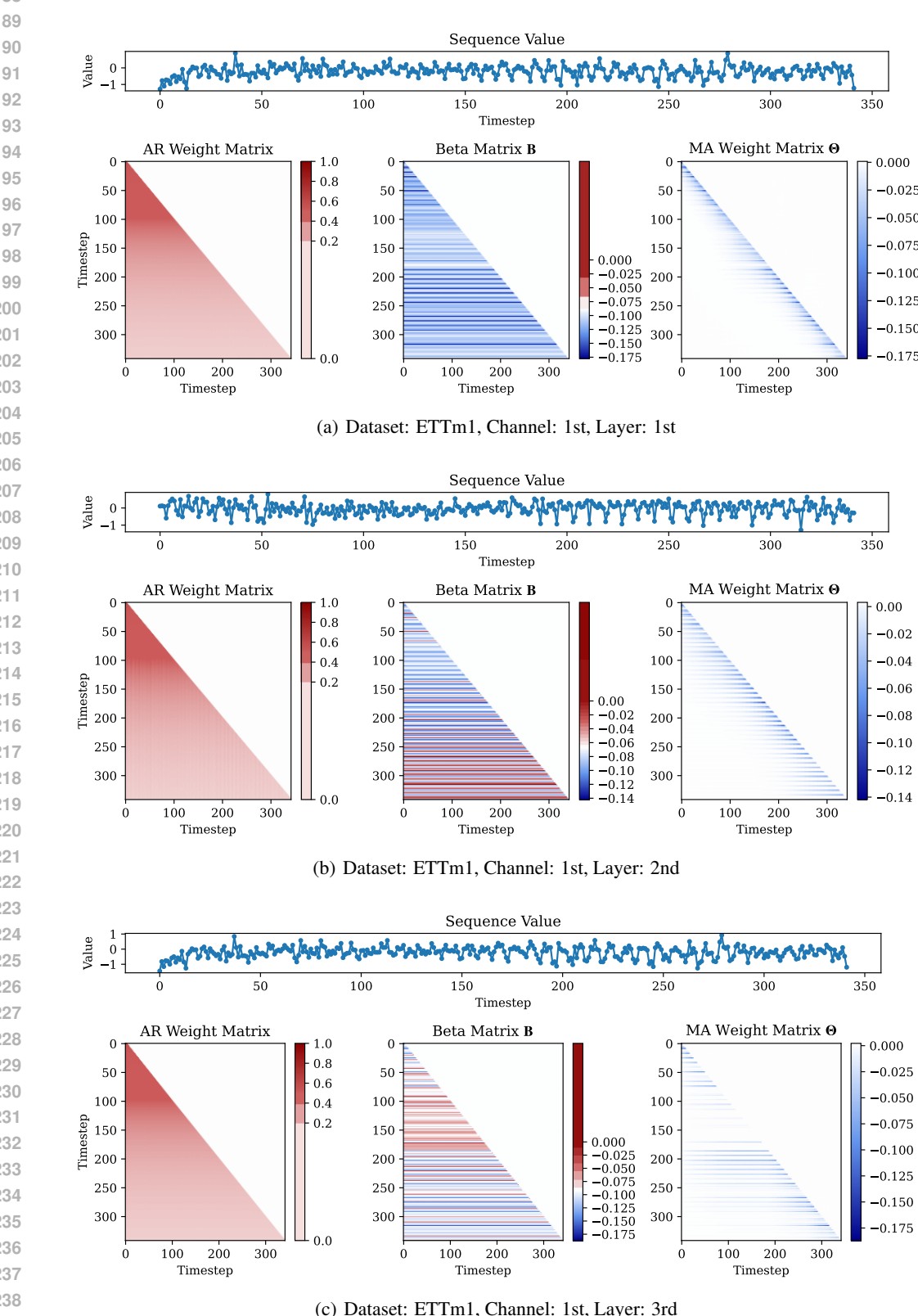

Figure 10: Visualization of the ARMA attention weights of the first input channel for the first test set data point in the ETTm1 dataset ($L_I = 4096$, $L_P = 12$).

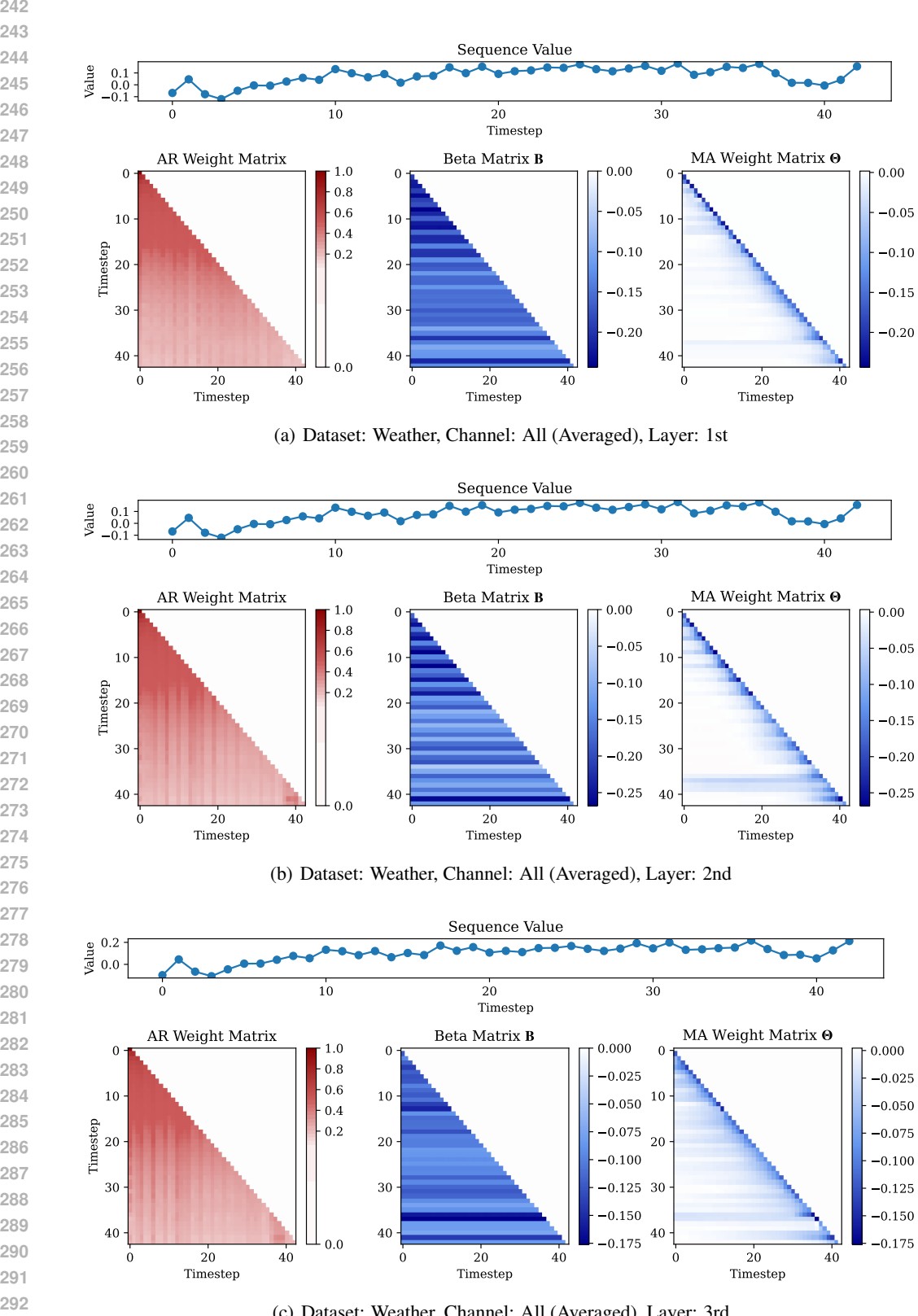

(a) Dataset: Weather, Channel: All (Averaged), Layer: 1st

(b) Dataset: Weather, Channel: All (Averaged), Layer: 2nd

(c) Dataset: Weather, Channel: All (Averaged), Layer: 3rd

Figure 11: Visualization of the ARMA attention weights for the first test set data point in the Weather dataset ($L_I = 4096$, $L_P = 96$).

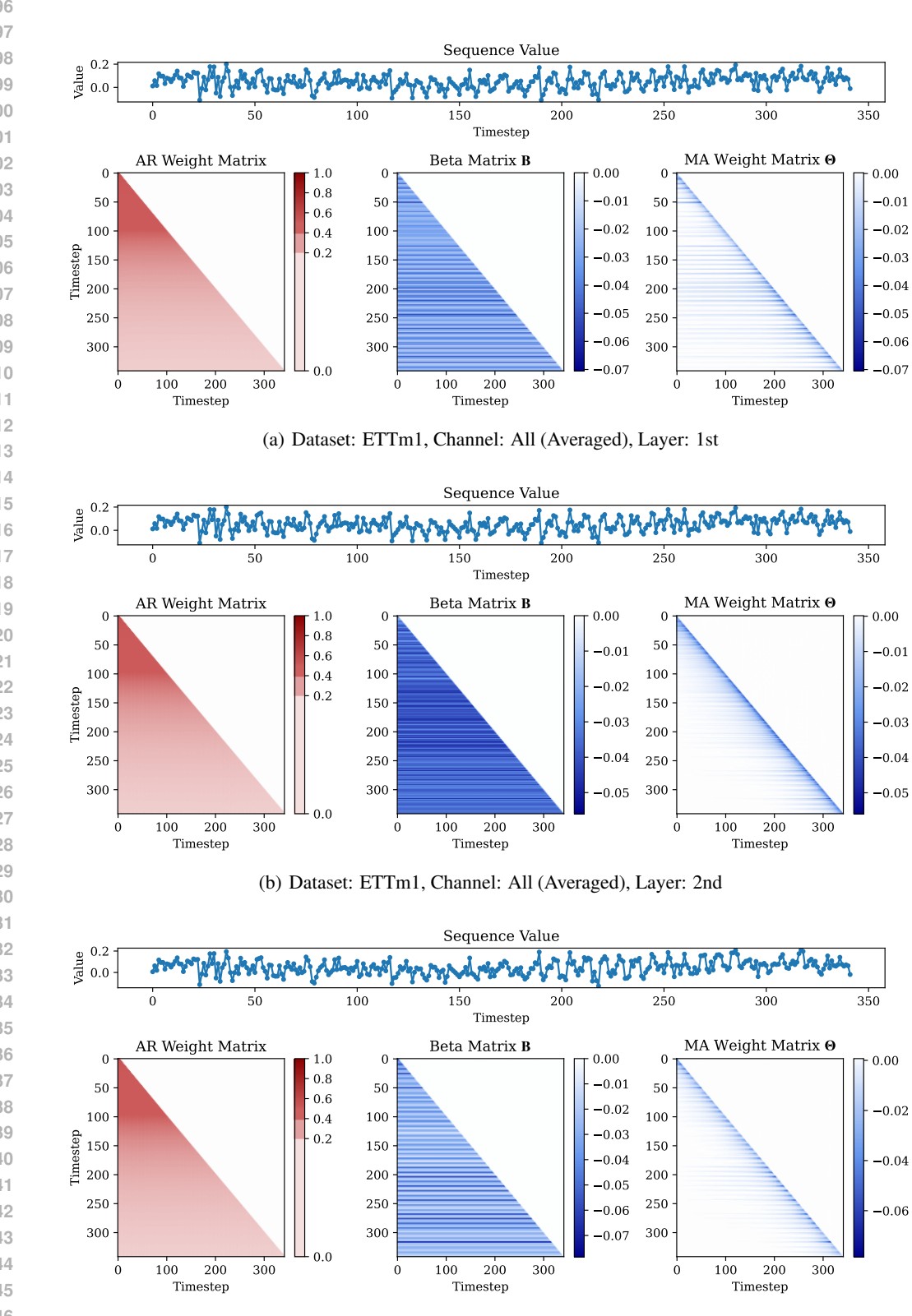

(a) Dataset: ETTm1, Channel: All (Averaged), Layer: 1st

(b) Dataset: ETTm1, Channel: All (Averaged), Layer: 2nd

(c) Dataset: ETTm1, Channel: All (Averaged), Layer: 3rd

Figure 12: Visualization of the ARMA attention weights for the first test set data point in the ETTm1 dataset ($L_I = 4096$, $L_P = 12$).

