# OpenReview forum: "Autoregressive Moving-average Attention Mechanism for Time Series Forecasting"
_ICLR.cc/2025/Conference — ICLR 2025 Conference Withdrawn Submission_

### Official Review · Reviewer_Raj3 · 2024-10-27

**Soundness:** 3
**Presentation:** 3
**Contribution:** 3
**Rating:** 6
**Confidence:** 4

**Summary:**

The author introduces a moving average term into the autoregressive attention model for linear attention mechanisms, indirectly computed using a linear RNN to collect all keys and values at once.

**Strengths:**

The idea is clever, and experimentally, all the method's aspects seem well analyzed: from the added computational load perspective to the combination of the MA term with different attention mechanisms. The supplementary material offers a complete experimental evaluation.

**Weaknesses:**

Some issues from which I would like clarification/further insights:

- You stress the importance of the MA in capturing short-term effects and letting the attention model include long-term effects. Wouldn't it be better to exploit the signal's spectral properties to address that? Two recent approaches that consider this idea (among many others in previous years) are Fredformer (https://arxiv.org/abs/2406.09009) and TSLANet: (https://arxiv.org/pdf/2404.08472). I think these two papers should be included in the comparison.

- You state that error accumulation is avoided in section 2.2. by jointly predicting signal periods of length $L_P$ (instead purely next token prediction of the signal). Is that a reasonable assumption?  When you need to forecast a signal with length above $L_P$, you have to feed back your estimate to the first $L_P$ terms back to the decoder to estimate the next signal values...

- Why do you only show your results using MSE? A quantile loss is definitely more descriptive and reliable, and it should be included.

- As you say, you have not explored combining the channel-independent ARMA Transformer with multivariate forecasting models to improve forecasting. Namely, you fit one model per channel. Have you done the same for the rest of the baselines? I would say this is not fair if the baselines can easily integrate the channels into the same model and exploit interdependencies.

**Questions:**

See my comment above

---

### Official Review · Reviewer_N3MS · 2024-11-03

**Soundness:** 3
**Presentation:** 1
**Contribution:** 3
**Rating:** 3
**Confidence:** 4

**Summary:**

The authors employ an autoregressive decoder only transformer architecture to time-series data. The autoregressive nature is supplemented with a moving average "corrector" component inspired by statistical temporal analysis methods. The authors show that their method outperforms other methods on short-range time predictions.

**Strengths:**

This work shows strong results against other models and applies an autoregressive decoder-only method, which I believe has more potential in time-series forecasting. The authors do a good job explaining other types of attention and make an attempt to cast their work relative to other types of attention. This context is both very helpful and gives the reader confidence, but the paper lacks some clarity to fully realize these comparisons (described later). In my opinion, the underlying work, results, and usage of the decoder-only architecture can be sufficient for publication here but not in its current state.

**Weaknesses:**

The paper suffers from a lack of clarity and detail.

1) Section 2.5 needs to be completely rewritten with more detail and explanation. This section is crucial to understanding the rest of the paper and deserves more space and detail. It is unclear to me how the MA term fits into all of this. While this section attempts to explain it, it is unclear. For example, "In an ARMA model, the MA term captures short-term fluctuations." This is the fits time MA is mentioned and the authors already say that MA is in ARMA, implying the reader should know the architecture a prior.
2) There is a lot of time-series work, and much of it uses longer time horizon predictions (as the authors mentions). However, it was not clear to me that this work focuses on shorter time horizons until far later in the paper. The authors need to make this more clear.
3) The authors need to motivate why predicting on shorter time-scales is more important since other time-series methods can predict further into the future.
4) Figure 3 looks important but is not mentioned or used in section 2.5. This seems like a misuse of an important figure.
5) Please align the vectors in Eq. 3
6) It is very hard to read the labels in Fig. 4
7) In general it is difficult to read some of the tables, the text is fairly small.

**Questions:**

I am concerned about the authors' claim that they mitigate error accumulation, and how the MA contribution accounts for short-term dynamics. However, this concern may be caused by confusion induced by the paper's language in section 2.5.

1) The authors state that the MA term captures short-term fluctuations. Generally, taking a moving average destroys short-term information as it filters high-frequency Fourier components. It is not even clear to me what "short-term" means: 1, 5, 10, ... ? Using a moving average will destroy information at a time-scale shorter than what I believe the authors call "short-term" but this vague language leaves the reader with relative adjectives and little quantitative grounding in time-scale.
2) The model describes $r$ as the sum of the MA term and a remaining error $\epsilon_t$. However, in the autoregressive rollout one does not have ground truth for future predictions and therefore cannot find  $\epsilon_t$ and $r$ reduces to the MA term, making $\epsilon_t=0$ for that time step. I don't understand how this mitigates the error accumulation.

---

### Official Review · Reviewer_VFii · 2024-11-03

**Soundness:** 3
**Presentation:** 3
**Contribution:** 2
**Rating:** 5
**Confidence:** 4

**Summary:**

This paper proposes an autoregressive moving-average (ARMA) attention module that can be integrated into linear attention for time series forecasting.

Contributions: The authors introduce the ARMA structure into existing autoregressive attention-based forecasting models. From a practical implementation perspective, the authors propose an indirect moving-average (ma) weight generation that maintains the complexity of the underlying attention-based models. It leads to 10 AR/ARMA models for which large-scale experiments are conducted with 12 datasets and 6 baselines. Ablation studies are also conducted.

**Strengths:**

- The paper is well-written and well-structured
- The authors provided large-scale experiments with ablation studies
- Combining ARMA and attention is simple yet effective which is positive in my opinion
- The indirect MA weight generation is interesting and seems efficient with the weight-sharing

**Weaknesses:**

*Performance*
- It seems that the authors did not conduct multiple runs of experiments with different random seeds. Hence, it reduces the significance of the results. While this is also the case in past works (PatchTST, Informer, Autoformer, etc.), I do believe that this is not good practice. In recent papers, e.g. in SAMformer [3],  the authors perform multiple runs which enables the statistical significance of the results and hence a better comparison.
- The authors propose a total of 10 models AR/ARMA among which only one (Lin Att + ARMA) is superior to the baselines considered and I cannot tell whether the improvement is significant.

*Computational cost and efficiency*
- I think that the autoregressive framework will not be efficient in long-term forecasting because of the H iterations needed to predict at horizon H. Hence, only comparing computational costs in terms of FLOPs does not take into account the entire process. Could the author compare the efficiency of their models also in terms of total training/inference (GPU) time?
- The authors mention in future work the extension to multivariate forecasting. I believe that the proposed models will be even less efficient in multivariate forecasting because of the autoregressive framework. Could the authors elaborate on that?

*Missing baselines*
- I am surprised the authors did not compare their approach to the original non-deep learning ARMA model [1]. Could the author compare to them?
- The authors did not compare their results to two strong recent models: TSMixer [2] and SAMformer [3]. These models are relevant given their performance and low computation cost, especially for SAMformer which is a transformer-based model. Could the author compare their approach to those models and mention them in the related work?

*Anonymity*

The MIT Licence of the code (available at https://anonymous.4open.science/r/ARMA-attention-3437/LICENSE) contains the name of one of the authors. It should be noted that I am not familiar with it and simply mention this as a warning to advise the authors to remove the Licence in their anonymous code.

**Questions:**

*Related to weaknesses*
- Could the authors perform multiple runs to obtain information on the statistical significance of the results?
- Could the author compare to the original non-deep learning ARMA models [1]?
- Could the authors perform a comparison with the two missing baselines TSMixer [2] and SAMformer [3] and mention them in the related work?
- Could the author compare the efficiency of their models also in terms of total training/inference (GPU) time?

*Potential typos*
- l269: the footnote is separated amount two pages. I think it would be better to keep it together for readability



**References**

[1] Box et al. *Some recent advances in forecasting and control*. Journal of the Royal Statistical Society, 1974

[2] Chen et al. * TSMixer: An all-MLP architecture for time series forecasting*. TMLR, 2023

[3] Ilbert et al. *SAMformer: Unlocking the Potential of Transformers in Time Series Forecasting with Sharpness-Aware Minimization and Channel-Wise Attention*. ICML, 2024

---

### Official Review · Reviewer_LnME · 2024-11-03

**Soundness:** 2
**Presentation:** 2
**Contribution:** 2
**Rating:** 3
**Confidence:** 3

**Summary:**

This paper introduces an Autoregressive (AR) Moving-Average (MA) attention structure that enhances the ability of linear attention mechanisms to capture both long-range and local temporal patterns in time series forecasting. By incorporating the ARMA structure into autoregressive attention mechanisms and using indirect MA weight generation, the proposed model improves performance on time series forecasting tasks.

**Strengths:**

1. The paper introduces the ARMA model from statistics into the autoregressive attention mechanism. This structure is classic in time series analysis and can effectively handle and decouple long-term and short-term effects.
2. The ARMA attention mechanism captures short-term effects through the MA term, allowing the AR term to focus more on long-term and periodic patterns, thus balancing long-term and short-term dependencies.
3. The ARMA attention mechanism proposed in the paper introduces the MA term while maintaining the same time complexity (O(N)) and parameter scale as the underlying efficient attention model.

**Weaknesses:**

1. The main advantage of a decoder-only model is that it allows for training just one model for inference at various lengths. Although there may be some cumulative error, it can still achieve state-of-the-art performance. The article's approach of "treating one-step prediction as the complete forecast" loses this advantage, as it requires training a separate model for each length.
2. The prediction length of {12, .., 96} as the main experimental setting is also unsatisfactory. The so-called fair comparison is primarily due to the fact that the input length must change according to the output length, which affects the model's applicability.
3. I don't like the large amounts of color blocks used in the experimental table.

**Questions:**

1. Sensitivity of parameters not analyzed.
2. Why is each row of matrix B in the visualization basically uniform?

---

### Author Response · Authors · 2024-12-03

Thank you to all reviewers for your valuable feedback. Below, we provide concise responses to a few common questions:

### 1. How does the AR/ARMA Transformer structure in the paper reduce the impact of error accumulation typical in autoregressive-based TSF models? Doesn’t long-term forecasting still require iterative inference?

As discussed in the paper, we mitigate error accumulation by setting the output length \( L_P \) equal to the patching size, allowing the model’s final token output to directly cover the forecasting horizon. This approach avoids iterative predictions. Please note, we do not consider this a structural improvement but rather an effective usage of the autoregressive framework.

If there is any confusion regarding our model’s structure, it might help to view the AR/ARMA Transformer directly as a PatchTST-like encoder-based Transformer, equipped with causal masking and a next-token prediction loss. When using softmax attention, our AR structure simply adds a mask and a loss with the other parts being identical to PatchTST. Readers can think of these additions as a form of regularization applied to encoder-based TSF models for handling variable-length inputs. With this perspective, the structure should no longer appear unnatural.

Additionally, the extensible output nature of autoregressive models means our model can indeed generate predictions beyond \( L_P \). The next-token prediction loss ensures these extended outputs outperform encoder-based models without such a design. However, since prior TSF structures did not intentionally explore this direction, we chose not to highlight this advantage in our experiments. To maintain fairness and consistency with baseline comparisons, we adhered to the current experimental setup.

### 2. Why did we choose output lengths of {12, 24, 48, 96} in the main experiments instead of the more common {96, 192, 336, 720} used in long-term time series forecasting?

Existing TSF models often struggle with handling longer lookback windows effectively, with the maximum lookback typically capped at 720. Extending the lookback window beyond this point often leads to significant overfitting on many datasets. In our case, due to the use of \( L_P \)-length patching tokenization in the AR structure, setting \( L_I = 720 \) for \( L_P = 720 \) would result in the input containing only a single token, reducing the model to an simple MLP. This would fail to showcase the benefits of the proposed ARMA attention mechanism. While the model can still learn in this setting, we do not recommend configurations that fail to leverage ARMA attention's advantages.

Moreover, prior models tend to experience performance drops with longer \( L_I \) and \( L_P \), but perform well in shorter settings. Thus, comparing our model with baselines in their optimal shorter range actually better highlights the improvements brought by the AR/ARMA Transformer.

Outside the main experiments, we also conducted tests with extended \( L_I \) to 4096 and \( L_P \) up to 720. These experiments demonstrated even more significant advantages for the AR/ARMA Transformer in such settings. Therefore, concerns that the chosen {12, 24, 48, 96} settings might unfairly favor the AR/ARMA Transformer are actually unfounded. On the contrary, we intentionally compared the models in settings where ARMA attention's theoretical advantages were less evident ({12, 24, 48, 96}), further emphasizing its significance.

Given the feedback from several reviewers, we plan to revise the experimental design in the next version to present these findings more clearly.

### 3. Why do we use the MA term, and why does it effectively decouple short-term effects?

In NLP data, short-term effects can be captured using techniques like local attention or exponential decay in gated linear attention. Exponentially decaying weights help increase locality, which is beneficial for modeling short-term effects. However, while AR weights are naturally suited for capturing long-term effects, adding exponential decay tends to suppress these long-term patterns, especially when some of the long-term compnents are stable. In NLP, long-term effects usually occur sporadically at isolated time points, often in scenarios such as token retrieval.

In contrast, TSF data frequently exhibit stable seasonal patterns that span the entire temporal dimension. Using gating mechanisms to emphasize locality can suppress these stable seasonal patterns, making the AR weights less effective. By incorporating MA weights to independently capture local effects rather than suppressing distant AR weights, we allow the AR component to focus on modeling stable seasonality and occasional long-term effects—areas where it excels.

This decoupling enables the AR part to retain its strength in capturing stable and long-term patterns, while the MA part specializes in modeling short-term effects, leading to a more effective and balanced approach.

---

> ### Author Response · Authors · 2024-12-03
>
> ### 4. SOTA performance of AR/ARMA attention and the possibility of further comparisons
>
> First, we want to clarify our primary goal: to demonstrate how the MA term can be incorporated into AR attention effectively. Ideally, we would have used an well-established, SOTA-level AR attention-based TSF model as our baseline to highlight the significance of adding the MA term. Unfortunately, the TSF field lacks high-performing AR-based structures. As a result, we had to first introduce a feasible AR attention structure comparable to SOTA and then focus on the key contribution of this paper.
>
> Many reviewers have focused on the AR attention structure itself, which was not our intended emphasis. What we truly aim to highlight is the significant improvement ARMA attention offers over AR attention. While ARMA attention indeed achieves SOTA performance in our experiments, we believe its value is evident even if it didn’t fully reach SOTA when we wrote this paper. The improvement over AR attention alone provides meaningful insights.
>
> From the feedback, it seems that introducing the AR structure, which is not novel, may have diverted attention from the key contribution: the comparison between ARMA and AR attention. We will refine our presentation in future versions to avoid this misunderstanding.
>
> Regarding ARMA attention's SOTA performance, our code is fully open-source, and we welcome others to reproduce the results. If anyone identifies a better AR structure setting during reproduction, we would be eager to discuss about it further. A more robust AR baseline for TSF would better underscore the value of ARMA attention.

---

### Note · Authors · 2024-12-03

I have read and agree with the venue's withdrawal policy on behalf of myself and my co-authors.